# Integrating genomic and multiomic data for *Angelica sinensis* provides insights into the evolution and biosynthesis of pharmaceutically bioactive compounds

Shiming Li[1,2,3,8], Tsan-Yu Chiu [2,8], Xin Jin[2,8], Dong Cao[1,3], Meng Xu [2], Mingzhi Zhu[4], Qi Zhou[2], Chun Liu[2], Yuan Zong[1,3], Shujie Wang[2], Kang Yu[2], Feng Zhang[2], Mingzhou Bai[2,5], Guangrui Liu[1,3], Yunlong Liang[1,3], Chi Zhang[2], Henrik Toft Simonsen[5,6], Jian Zhao [4✉], Baolong Liu [1,3✉] & Shancen Zhao [2,7✉]

*Angelica sinensis* roots (Angelica roots) are rich in many bioactive compounds, including phthalides, coumarins, lignans, and terpenoids. However, the molecular bases for their biosynthesis are still poorly understood. Here, an improved chromosome-scale genome for *A. sinensis* var. Qinggui1 is reported, with a size of 2.16 Gb, contig N50 of 4.96 Mb and scaffold N50 of 198.27 Mb, covering 99.8% of the estimated genome. Additionally, by integrating genome sequencing, metabolomic profiling, and transcriptome analysis of normally growing and early-flowering Angelica roots that exhibit dramatically different metabolite profiles, the pathways and critical metabolic genes for the biosynthesis of these major bioactive components in Angelica roots have been deciphered. Multiomic analyses have also revealed the evolution and regulation of key metabolic genes for the biosynthesis of pharmaceutically bioactive components; in particular, *TPSs* for terpenoid volatiles, *ACCs* for malonyl CoA, *PKSs* for phthalide, and *PTs* for coumarin biosynthesis were expanded in the *A. sinensis* genome. These findings provide new insights into the biosynthesis of pharmaceutically important compounds in Angelica roots for exploration of synthetic biology and genetic improvement of herbal quality.

[1] Qinghai Province Key Laboratory of Crop Molecular Breeding, Key Laboratory of Adaptation and Evolution of Plateau Biota, Northwest Institute of Plateau Biology, Chinese Academy of Sciences, 810008 Xining, Qinghai, China. [2] BGI-Shenzhen, 518083 Shenzhen, Guangdong, China. [3] The Innovative Academy of Seed Design, Chinese Academy of Sciences, 810008 Xining, Qinghai, China. [4] National Research Center of Engineering and Technology for Utilization of Botanical Functional Ingredients, Hunan Agricultural University, 410128 Changsha, Hunan, China. [5] Department of Biotechnology and Biomedicine, The Technical University of Denmark, 2800 Kongens, Lyngby, Denmark. [6] Laboratory of Plant Biotechnology, Université Jean Monnet, 23 Rue du Dr Michelon, 42000 Saint-Etienne, France. [7] Beijing Life Science Academy, 102200 Beijing, China. [8] These authors contributed equally: Shiming Li, Tsan-Yu Chiu, Xin Jin. ✉email: jzhao2@qq.com; blliu@nwipb.cas.cn; zhaoshancen@genomics.cn

*A*ngelica sinensis (Oliv.) Diels (2n = 22), known as Dang-gui in China, belongs to the Apiaceae family and is one of the most popular traditional herbal medicines[1]. The medicinal applications of its dried roots are officially documented in Shennong Bencao Jing, dated back to 200–300 AD[2]. These recipes are issued to treat various illnesses, such as replenishing blood and treating abnormal or painful menstruation, uterine bleeding, and other diseases affecting women, which has led to its common name "female ginseng"[3,4]. In addition to its common usage as a medicine in Asia, *A. sinensis* has also been widely marketed in Europe and America as a dietary supplement because of its pleasant flavors[5].

Over 140 chemical compounds have been isolated and identified from dried Angelica roots[2,5]. The bioactive constituents in the root of *A. sinensis* include nonvolatile and volatile compounds. Terpenoid volatiles and essential oils, including phthalides, such as ligustilide, butylidene phthalide, and butyl phthalide, are the major bioactive compounds[6]. While butylidene phthalide, butyl phthalide, and senkyunolide A have been used to treat hypoglycemic and ischemic stroke diseases and for vasorelaxing, neuroprotective, and antitumor applications[7–10], organic acids such as ferulic acid have been reported to provide anti-inflammatory and antioxidant effects[11,12]. Despite their strong bioactivity and medicinal importance, the biosynthesis pathways and genes for the production of these phthalides in *A. sinensis* and other plants remain unknown. The identification of these pharmaceutically important bioactive ingredients and dissection of their biosynthesis pathway and genes are critical for improving Angelica medicinal value and the utilization of germplasm resources, eventually leading to the genetic improvement of Angelica medicinal quality.

Recently, a genome of *A. sinensis* was reported (collected from Gansu Province, China, and abbreviated as *A. sinensis* var. Gansu (GS) in this article), and genome evolution and coumarin biosynthesis were analyzed[13]. Using the 2.37 Gb assembled genome, the authors found that *A. sinensis* diverged from *Panax ginseng* (Araliaceae) ~49.5 million years ago (MYA), which diverged earlier from *Coriandrum sativum* and *Apium graveolens* at ~18.4 MYA, and thus it is more primitive. The authors also identified the genes involved in coumarin biosynthesis[13]. As the most important bioactive ingredients in Angelica roots, phthalides are responsible for the bioactivity and pharmacological properties of *A. sinensis*, such as anti-asthma effects, anti-convulsant effects, inhibition of platelet aggregation, and enhancement of blood flow[14]. However, the metabolic pathways and enzymes for the biosynthesis and transformation into different phthalides are still poorly understood[15,16].

*A. sinensis* naturally grows in cool, high-altitude mountainous regions between 2500 m and 3000 m above sea level and is cultivated in northwest provinces of China, such as Qinghai and Gansu provinces[17]. *A. sinensis* has a normal growth cycle of 3 years, with seedlings raised in the first year, medicinal root formation in the second year, and bolting and flowering in the third year. However, 20–30% of *A. sinensis* plants can bolt and flower in the second year due to germplasm heterozygosity[18]. Early bolting and flowering have multiple negative effects on Angelica roots' medicinal qualities due to fewer secondary metabolites, especially a significant reduction in essential oil components mainly composed of phthalides[15]. Six phthalides, namely, ligustilide, butylphthalide, butylidenephalide, senkyunolide H, senkyunolide I, and senkyunolide A, in Angelica roots decreased obviously in early-flowering plants, which significantly deteriorated the medicinal value of Angelica roots[15]. However, the dynamic changes in these important bioactive phthalides in early-flowering *A. sinensis* and the underlying mechanisms are largely unknown.

*A. sinensis* var. Qinggui1 is a widely cultivated and commonly used variety in Qinghai Province (referred to as *A. sinensis* (QH) in this article). We present a chromosome-scale genome assembly of *A. sinensis* var. Qinggui1 using three technologies: single-molecule real-time (SMRT) sequencing, chromosome conformation capture (Hi-C) sequencing, and next-generation sequencing (NGS). With this genome, we further investigate the evolutionary relationships between *A. sinensis* and other species in the Apiaceae family, as well as the content and variation of the major bioactive natural products, phthalides, in Angelica roots before and after early flowering by integrating transcriptome and metabolite profiling analyses. An in-depth analysis of related gene families provides insights for further elucidating the biosynthesis and regulation of medicinal active ingredients in *A. sinensis*.

## Results

**Genome assembly and annotation.** The widely cultivated *A. sinensis* cultivar Qinggui1 was selected for genome sequencing (Fig. 1a). We generated a total of 376.4 Gb Single Molecule Real-Time (PacBio SMRT) sequences and 60.8 Gb paired HiSeq reads (PE150), along with 325.0 Gb effective chromosome conformation capture (Hi-C) reads (Table S1). The assembly was initialized by PacBio SMRT sequences, which were corrected with high-quality paired HiSeq reads. A genome size of 2.16 Gb was obtained after the final assembly. The Hi-C interaction matrices showed a distinct separation pattern of 11 blocks that could be used to cluster and orient the contigs and anchor them to 11 chromosomes (Fig. 1b and Tables 1 and S2). The size of the genome that we assembled was similar to the size estimated by flow cytometry[13]. Mapping the short reads back to the assembly led to a correction of 29,533 single-base errors and 9426 small Indels. The identification of 1,588,740 heterozygous SNPs showed a low level of heterozygosity in this self-fertilized plant. Evaluation by the Benchmarking Universal Single-Copy Orthologs (BUSCO) method[19,20] showed >99% completeness of the genome (Table S3). These results confirm a high-quality genome assembly. Please refer to Table 1 and Data availability for detailed information on the genome assembly.

Approximately 80.24% of the assembly (1.66 Gb) was identified to be repetitive sequences, which was higher than estimates in another Apiaceae family member, coriander (70.59%) (Fig. 1c, Table S4). Long terminal repeats (LTRs), primarily consisting of *Gypsy* and *Copia* subtypes, were most abundant. The other repeats were categorized as DNA transposons (3.65%), long interspersed nuclear elements (LINEs; 1.26%), short interspersed nuclear elements (SINEs, 0.03%), and uncharacterized repeats (19.77%) (Table S5).

We predicted a total of 41,040 protein-coding genes (Table S6) using ab initio methods, protein homology, and RNA-seq reads from different tissues. Of them, 98.3% were mapped to the chromosomes, and most were distributed in the terminal regions (Fig. 1c). Using the iTAK pipeline[21], we predicted 2,996 transcription factor (TF) genes in the *A. sinensis* genome. The top five TF families were MYB/MYB-related (209), AP2/ERF-ERF (172), bHLH (166), C2H2 (154), and NAC (135). Compared with those in other Apiaceae plants, GeBP, HSF, GARP-G2-like, C2C2-GATA, C2C2-Co-like, HB-WOX, and Trihelix families were expanded whereas C2C2-YABBY, B3-ARF, and GRAS genes dramatically decreased in *A. sinensis* (Fig. S1). The genome that we assembled in this study included more TF genes in most TF families than that in the published *A. sinensis* (GS) genome (Fig. S1).

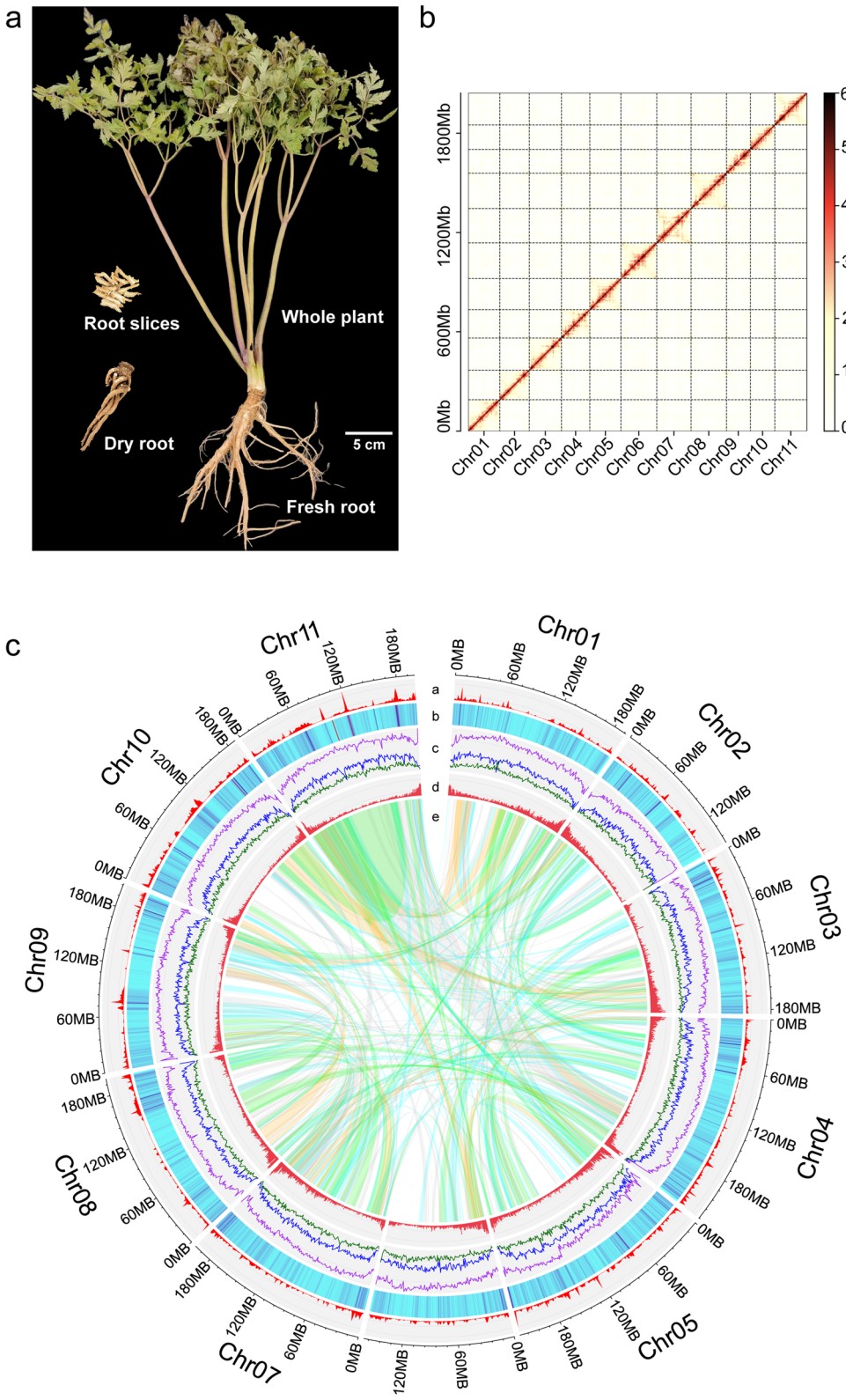

**Fig. 1 Overview of the *A. sinensis* (QH) genome. a** Morphology of the sequenced plant. **b** Hi-C map of chromosomes. **c** a-b. SNP and indel density and distribution identified between *A. sinensis* (GS) and *A. sinensis* (QH); **c** Density and distribution of LTR retrotransposons (purple: LTR; blue: Copia-type; dark green: Gypsy-type); **d** Gene density and distribution; **e** Colinear gene pairs within the genome. The colors of linking lines indicate the number of one-to-one gene pairs in the collinearity blocks: ≥40, green: ≥20, blue: ≥10, gray: ≥5. This figure was prepared by using shinyCircos[110].

**Table 1 Summary of the *A. sinensis* (QH) genome assembly and annotation.**

| Genome assembly | Total length of contigs (bp) | | 2,156,060,282 |
|---|---|---|---|
| | GC content (%) | | 35.60 |
| | N50 length (contigs) (bp) | | 4,960,951 |
| | Longest contig (bp) | | 20,518,691 |
| | Total length of scaffolds (bp) | | 2,155,436,444 |
| | N50 length (scaffolds) (bp) | | 198,273,161 |
| | Longest scaffold (bp) | | 228,427,485 |
| Transposable elements | Annotation | Percentage | Total length (bp) |
| | Retrotransposons | 59.24 | 1,276,667,829 |
| | DNA transposons | 3.83 | 82,512,686 |
| | Others | 17.17 | 370,118,161 |
| | Total | 80.24 | 1,729,298,676 |
| Protein-coding genes | Predicted high-confidence genes | | 41,040 |
| | Average gene length (bp) | | 3951.49 |
| | Average coding sequence length (bp) | | 1126.57 |
| | Average exon length (bp) | | 240.09 |
| | Average intron length (bp) | | 765.08 |
| | Functionally annotated | | 39,300 |

**Genome evolution of typical herbaceous angiosperm plants and whole-genome duplication in the Apiaceae family**. Despite the increasing number of sequenced genomes of medicinal plants, systematic studies of their evolutionary relationships are relatively scarce. To explore the phylogenetic position of *A. sinensis* in the Apiaceae family and its evolutionary relations with other species, we selected typical representative families/orders and medicinal plant species of rosids and asterids according to the Angiosperm Phylogeny Group classification (APG V4) classification system[22] and constructed a phylogenetic tree using one-to-one homologous gene families. These 20 representative angiosperms included 12 well-known medicinal plant species (Table S7) from 14 families and 12 orders, representing the major botanical taxonomic groups of core eudicots.

Among these species, *Vitis vinifera* was chosen for its important evolutionary position and its wide use as a model and basal plant for plant evolutionary research[23]. *Arabidopsis thaliana* and *Solanum lycopersicum* are well-studied model eudicot plants[24,25]. *Theobroma cacao* and *Camellia sinensis* are two of the most important beverage crops and are rich in secondary metabolites such as caffeine[26,27]. *C. sinensis* is also one of the basal species of asterid plants[27]. *Populus trichocarpa* was selected as a model plant for the study of lignin biosynthesis and phenylpropanoid metabolism[28], which is also one of the most important metabolic pathways in *A. sinensis* related to the bioactive metabolites of ferulic acid, lignans, and coumarins. *Cannabis sativa* is one of the most valuable agriculturally important crops in nature and is also used to produce well-known drugs - tetrahydrocannabinol (THC) and cannabidiol (CBD)[29]. *Ophiorrhiza pumila*, belonging to the family Rubiaceae, is an important herbaceous medicinal plant and can accumulate camptothecin (CPT)[30]. *Scutellaria baicalensis, Salvia miltiorrhiza, Taraxacum mongolicum, Artemisia annua, Lonicera japonica, Panax notoginseng, Panax ginseng, Angelica sinensis,* and snapdragon (*Antirrhinum majus* L.) are widely used as traditional Chinese medicines with thousands of years of history in China. In addition, we also included *Daucus carota, Apium graveolens,* and *Coriandrum sativum*, which are important members of the Apiaceae family, to examine the evolutionary relationships within the family and the evolutionary status of *A. sinensis*.

We identified a total of 2133 one-to-one orthologous gene families shared by all the species (Fig. S2). Using these orthologs, we constructed a phylogenetic tree by the concatenation method. As expected, the topology of the tree was consistent with the APG V4 classification. In the Apiales order, Araliaceae was grouped with Apiaceae, and Araliaceae was considered to be the ancestral family. Divergence time estimates showed that these two families separated around 58 MYA. Within the Apiaceae family, *A. graveolens* and *D. carota* diverged approximately 23 MYA, which is much earlier than the divergence of *A. sinensis* (QH) and its sister clade *C. sativum* (12 MYA) (Fig. 2a).

To further investigate the evolutionary relationships among Apiaceae species, we clustered approximately 91.3% (206,682) of the genes from five Apiaceae species and one outgroup species (*P. notoginseng*) into 29,108 orthologous groups and extracted 3189 single-copy genes (Table S8). We constructed a phylogenetic tree based on the concatenated sequence alignment of these single-copy gene families (Fig. 2b). *C. sativum* showed the most marked gene expansion. *A. sinensis* (QH) and *A. sinensis* (GS) were clustered together and *C. sativum* was their closest relative. *A. sinensis* (QH) had more expanded and fewer contracted gene families than *A. sinensis* (GS) (Fig. 2b).

We identified 3698 genes as members of significantly expanded gene families ($P < 0.01$) in *A. sinensis* (QH) and mapped them to the Kyoto Encyclopedia of Genes and Genomes (KEGG) pathways for functional enrichment analysis. We detected 33 significantly enriched pathways ($P < 0.05$), and the top enriched metabolic pathways included "Glycosphingolipid biosynthesis", "Zeatin biosynthesis", "Benzoxazinoid biosynthesis", "Oxidative phosphorylation", "Sesquiterpenoid and triterpenoid biosynthesis", "Biosynthesis of unsaturated fatty acids", "Selenocompound metabolism", and "Indole alkaloid biosynthesis" (Fig. 2c and Table S9). Some of the enriched KEGG pathways were involved in plant volatile biosynthesis, such as "Sesquiterpenoid and triterpenoid biosynthesis" and "Phenylpropanoid biosynthesis", which suggested that these genes may contribute to the adaptive phenotypic diversification of *A. sinensis* species.

Whole-genome duplications (WGDs) are widely recognized as a major source of species diversification in many eukaryotic lineages based on various lines of evidence[31]. To identify potential WGD events, we calculated the nucleotide divergence at fourfold synonymous third-codon transversion positions (4dTv) and the synonymous substitution rates (Ks) for collinear gene pairs within each species. In addition to the five members of the Apiaceae family, namely, *D. carota, A. graveolens, C. sativum, A. sinensis* (GS), and *A. sinensis* (QH), we also included the model plant *V. vinifera* in our study.

The intragenomic paralogous genes of the Apiaceae species exhibit three distinct peaks in their 4dTv distributions (Fig. 2d). The last peak (γ), shared with *V. vinifera*, signifies an ancient Whole Genome Triplication (WGT) event common to all eudicot plants. The first two peaks indicate two recent lineage-specific Whole Genome Duplication (WGD) events that took place prior to the divergence of the family members within the Apiaceae family. This observation aligns with a previous study which suggested that *A. sinensis* has undergone three polyploidy events[13]. By comparing the peak positions across species, we inferred a sequence of WGD events: *A. sinensis* experienced the most recent event, followed by *C. sativum* and then *A. graveolens*. This sequence corroborates our phylogenetic tree and divergence time estimates, thereby enhancing the consistency of our findings.

Ks values of homologous genes from different genomes can be used to estimate the time of species divergence[32]. In this study, we compared the Ks peak values within each species and identified two distinct peaks at Ks 0.5 and 1.0, corresponding to two WGD events (Fig. 2e). The peak positions of A. sinensis (QH) and A. sinensis (GS) were nearly identical (see Table S10 for complete peak values), suggesting similar evolutionary histories for these

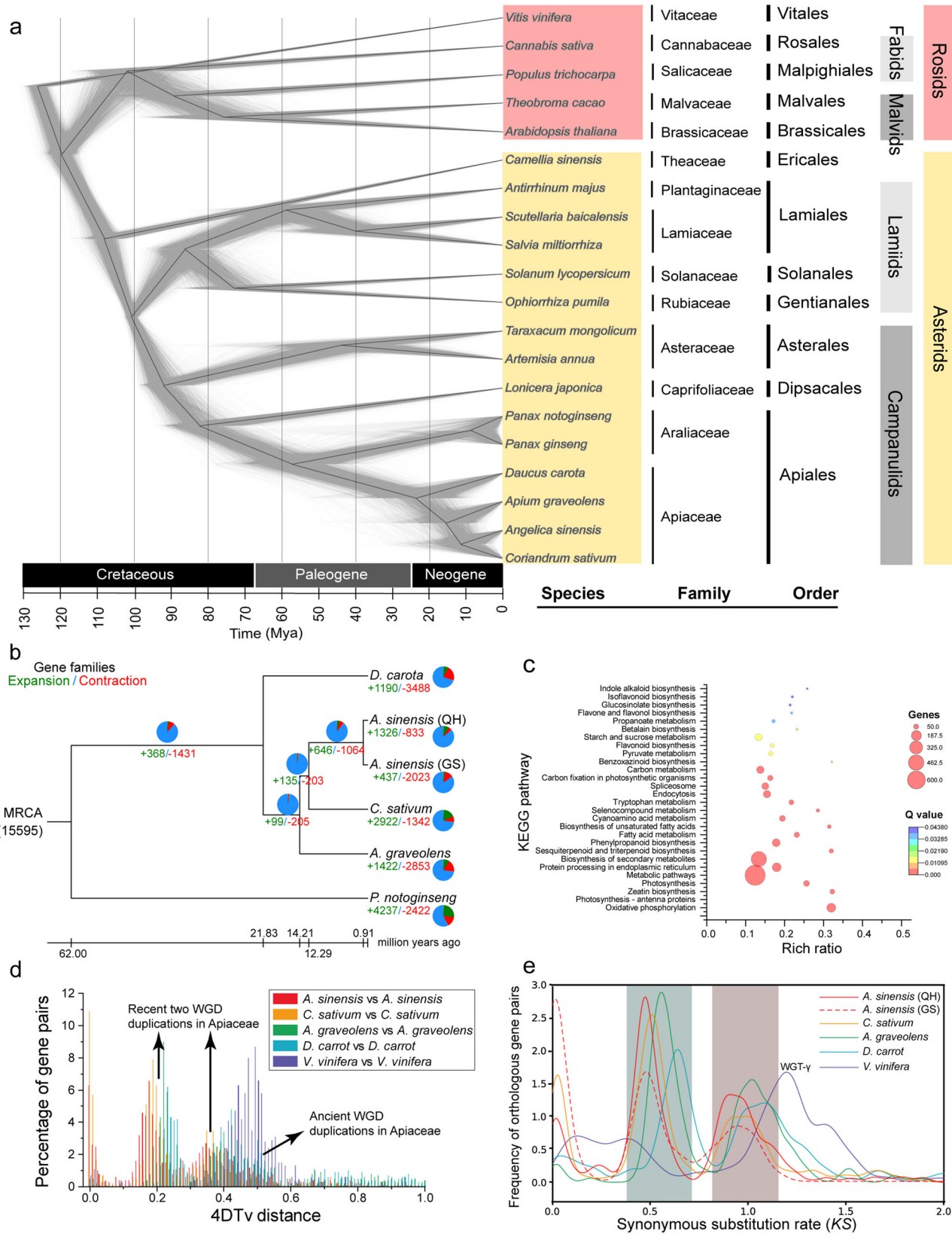

two varieties. However, the peak at around 1.7 is not evident, likely due to the loss or divergence of ancient duplicate genes following the earliest WGD event. The order of the peak values aligned with the phylogenetic relationships of carrot, celery, coriander, and Angelica. This implied that the order of WGD events happened in these species was carrot, celery, coriander, and Angelica which was also consistent with the previous 4dTV analysis.

**Fig. 2 Evolutionary analysis of representative angiosperm species and WGD events in the Apiaceae family. a** Molecular phylogenetic tree of 20 representative angiosperm species constructed using 2133 concatenated conserved protein sequences by the ML and BI methods. **b** Phylogenetic tree of *A. sinensis* and other Apiaceae species, inferred by estimating divergence time using 3188 single-copy ortholog sequences. *P. notoginseng* was used as an outgroup. The numbers in green and red colors indicate gene family expansion and contraction compared with the most recent common ancestors, respectively. Estimated divergence times (MYA, million years ago) are indicated at each node. The Venn diagram shows the proportion of gene families under the unchanged (blue), expansion (red) and contraction (green) scenarios. **c** KEGG pathway enrichment analysis of expanded gene families in the *A. sinensis* (QH) genome. Only the enriched KEGG pathways with *p* values < 0.05 are displayed. **d** Distribution of 4DTv distances of syntenic orthologous genes of Apiaceae species. The black arrows mark the WGD events. **e** The $K_S$ distribution for orthologous gene pairs within Apiaceae species. *V. vinifera* was used as the model organism for evolutionary analysis. The shape of the curve and the position of the peak are almost identical between *A. sinensis* (QH) and *A. sinensis* (GS). The highlighted peak regions represent two WGD events.

**Comprehensive comparison between *A. sinensis* (QH) and *A. sinensis* (GS) genomes.** A total of 41,040 high-confidence genes were predicted, which is 2,163 fewer than the published genome annotation of 43,202 genes. To evaluate the integrity of the gene set, both gene sets were first compared using the same BUSCO version and parameters. A proportion of complete genes of 96.41% was found in *A. sinensis* (QH), while *A. sinensis* (GS) had only 88.10%. Second, common databases, including the InterproScan[33], Gene Ontology (GO)[34], KEGG[35], SwissProt[36], TrEMBL, KOG, and nonredundant protein NCBI databases, were used to functionally annotate these two gene sets. Approximately 95.76% of the genes were annotated in *A. sinensis* (QH), while only 90.38% were annotated in *A. sinensis* (GS). Third, Ortho-Finder (v2.5.4)[37] was used to cluster these two gene sets for further analysis. The percentage of genes in orthologous groups was 94.9% in *A. sinensis* (QH), while it was only 82.6% in *A. sinensis* (GS). The species-specific gene number was 2,111 in *A. sinensis* (QH) and 7,496 in *A. sinensis* (GS). In summary, we provided a better reference gene annotation for *A. sinensis* species.

The genomic differences between *A. sinensis* (QH) and *A. sinensis* (GS) were investigated. Highly collinear relationships were evident between these two genomes (Fig. 3a, b). A large inversion was also observed along homologous chromosomes Chr09 (*A. sinensis* (QH)) and chr04 (*A. sinensis* (GS)), which is highlighted by a red arrow in Fig. 3a and a red square in Fig. 4b. Good collinearity was found in this region between *A. sinensis* (QH) and *A. graveolens*, suggesting that *A. sinensis* (GS) had an assembly error in this region or that this is an inherent feature of the *A. sinensis* (GS) genome. Relatively good collinearity was observed at the genome level between *A. sinensis* and *A. graveolens*. Furthermore, reciprocal translocations were observed along chromosomes 05 and 07 in *A. sinensis* (QH), as well as along chromosomes 09, 11, and 10 in *A. graveolens* (Fig. 3b). This phenomenon was consistent between *A. sinensis* (GS) and *A. graveolens*, further confirming the occurrence of translocations between these chromosomes. The collinearities between *A. sinensis* (QH) and other species in Apiaceae are displayed in Fig. S3.

A total of 1.227 million SNPs and 242,250 Indels were detected in syntenic blocks between the two *A. sinensis* genomes. The distributions of SNPs and indels were similar but uneven across the whole genome (Fig. 1c). Most of the genetic variations were located in the intergenic regions. Of these, 38,862 SNPs and 8887 indels were located in the coding regions, affecting 9,547 and 5,125 genes, respectively. Within coding regions, 909 genetic variations (affecting 686 genes) were annotated as having a strong effect on gene function, with frameshifts or changes at the start or stop codon (Supplementary Data 1). These genes were not evenly distributed across the whole genome (Fig. 3c) and enriched in the KEGG pathways of biosynthesis of various secondary metabolites, such as Indole alkaloid, Betalain, Isoquinoline alkaloid and Sesquiterpenoid, and triterpenoid biosynthesis (Fig. 3d). The numbers of SNPs and indels were higher on chromosomes 10 and 11 than those on other chromosomes (Fig. 1c and Table S11).

**Metabolic landscape of important bioactive ingredients in Angelica roots.** To understand the biosynthesis of various bioactive components in Angelica roots, we conducted non-targeted metabolomics profiling on normally growing and early-flowering Angelica roots. More than 716 high-confidence meta-bolites were detected and identified, including 39 flavonoids, 12 terpenoids, 47 alkaloids, 74 phenolic acids, 10 phthalides, 31 coumarins, and 24 lignans (Supplementary Data 2), of which 299 compounds were determined as differential metabolites using univariate and multivariate statistical methods with the para-meters of FC ≥ 2 or ≤ 0.5 and VIP (variable importance in pro-jection) ≥1, including 145 upregulated and 154 downregulated metabolites.

The class of metabolites appeared to have completely different metabolic patterns in the Angelica roots between NG (normal growth) and EF (early flowering and bolting) samples. The Angelica roots in NG samples were rich in organic acids, amino acids and derivatives, saccharides and alcohols, and nucleotides and derivatives, while the Angelica roots in EF samples were rich in phenolic acids, LPC, LPE, coumarins, lignans, flavonols, and flavonoids (Fig. 4a). In particular, the differential production of these bioactive compounds in NG and EF Angelica roots showed that some phthalides and coumarins were more highly accumu-lated in NG roots than in EF roots, whereas most lignans accumulated at higher levels in EF roots than in NG roots (Fig. 4b). It demonstrated the higher medicinal value of NG roots than EF roots since these phthalides and coumarins displayed more important bioactivities in experimental and clinical studies.

Transcriptome analyses of these Angelica roots under different developmental conditions also unveiled the differentially expressed metabolic genes in their biosynthesis pathways in line with metabolomics data (Fig. 4c). The metabolic genes putatively involved in the biosynthesis of lignans and coumarins, both of which are derived from the phenylpropanoid pathway that often leads to the biosynthesis of well-known lignin and flavonoids, were upregulated in EF roots compared with NG roots (Fig. 4c). In contrast, most genes putatively involved in phthalide and coumarin biosynthesis were expressed at higher levels in NG roots than in EF roots, consistent with their higher pharmaceu-tical values (Fig. 4c).

Although the common shared metabolic enzymes and path-ways involving lignin, coumarins, lignans, and flavonoids are well known, the specific genes/enzymes involved in the production of many coumarins and lignans are poorly understood[13,38,39]. This new Angelica genome assembly provided more than 100 metabolic genes that encode all known enzyme homologs involved in the biosynthesis of coumarins and lignans (Supple-mentary Data 3). The phenylpropanoid pathway genes, including phenylalanine ammonia lyase (PAL), cinnamate 4-hydroxylase (C4H), 4-coumaroyl-CoA ligase (4CL), hydroxycinnamoyl-CoA

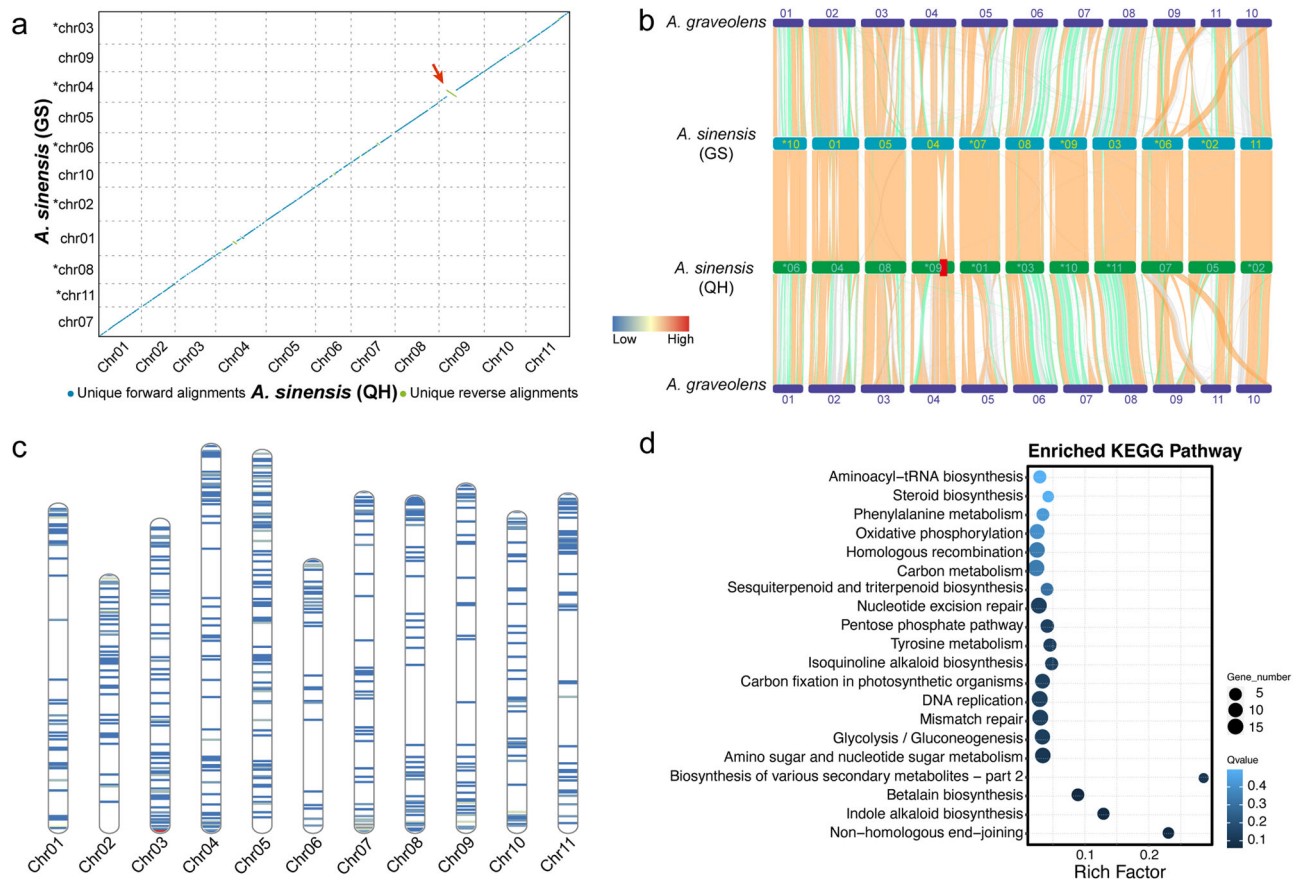

**Fig. 3 Genome collinearity analysis and gene family dynamics in *A. sinensis* and related species. a** Macrosynteny between *A. sinensis* (QH) and *A. sinensis* (GS) was verified using MUMmer[98] (version 4.0). Each dot represents a homologous block. Blue and green colors indicate different orientations of the sequences, while the red arrow refers to intrachromosomal inversions. The plot was generated using Dot (https://dot.sandbox.bio/). **b** Genome collinearity analysis among *A. sinensis* (QH), *A. sinensis* (GS), and *A. graveolens*. MCScanX[86] was used to identify collinear gene blocks among these three genomes. The red square highlights intrachromosomal inversions between *A. sinensis* (QH) and *A. sinensis* (GS). The color of linking lines indicates the number of one-to-one gene pairs in the collinearity blocks: orange (≥40), green (≥20), and gray (≥5). **c** The genome distribution of genes with strong functional effects between *A. sinensis* (QH) and *A. sinensis* (GS). **d** KEGG pathway enrichment analysis of genes with strong functional effects.

shikimate/quinate hydroxycinnamoyltransferase (HCT), caffeic acid *O*-methyltransferase (COMT), caffeoyl-CoA *O*-methyltransferase (CCoAOMT), etc., contributing to lignin biosynthesis via HCT and CCR genes, via dirigent protein (DIR), or via flavonoid synthesis by CHS and for coumarin biosynthesis from different products of 4CL by cinnamic acid 2'-hydroxylase (C2'H), p-coumarate 3-hydroxylase (C3'H) with HCT, or feruloyl-CoA hydroxylase (F6'H), were all assembled and annotated in our genome to provide insights on the biosynthesis of various pharmaceutically important products (Fig. 5a). Lignans have unique antitumor activities and reduce lifestyle-related diseases[40]. Lignans were also enriched in Angelica roots, particularly of EF status, in which a subset of biosynthesis genes and contents of lignans and derivatives were upregulated, including dirigent protein (DIR), pinoresinol-lariciresinol reductase (PLR), and secoisolariciresinol dehydrogenase (SIRD) for the biosynthesis of pinoresinol and lariciresinol, secoisolariciresinol, and matairesinol aglycones and their glycosides as products of UGT71/74 glycosyltransferses[40] (Fig. 5a).

Prenyltransferase (PT) catalyzes the prenylation of umbelliferone into linear or/and angular furanocoumarin biosynthesis[34,35]. PTs are involved in the biosynthesis of chlorophyll, vitamin E, heme, phylloquinone, and various secondary metabolites by prenyl modifications of chlorophyllide a/b, vitamin E, heme B, and many metabolites, such as 1,4-dihydroxyl-2-napthoic acid,

p-hydroxylbenzoic acid, flavonoids, phloroglucinol, homogentisate, and coumarins, with different prenyl donors, such as isoprenyl diphosphate, dimethylallyl diphosphate, and geranyl diphosphate (Fig. 5b). Despite the divergent functions of these PTs, they involved in coumarin biosynthesis that evolved most likely via convergent evolution since coumarins mainly occur in a few unrelated plant families, such as Fabaceae, Moraceae, Apiaceae and Rutaceae[34,35]. This finding is also supported by a previous study[19], which showed independent evolution of coumarin biosynthesis-related PTs in these families. Furthermore, these PTs that catalyze both linear (demethylsuberosin, e.g., PsPT1 and PcPT1) and angular (osthenol, e.g., PsPT2) furanocoumarin biosynthesis are clustered together in one clade for Apiaceae species (Fig. 5b), likely resulting from gene duplications followed by neofunctionalization and positive selection[38,41].

**Biosynthesis of phthalides and other terpenoid volatiles.** As two major pharmaceutically important components in Angelica roots, ligustilide and butylidenephthalide are generally regarded as essential contributors to the main medical functions of Angelica roots[42–45]. However, their biosynthesis pathways remain elusive. The oxidation or transfer of isoprenoids or condensation of malonyl CoAs with other acyl CoAs by type III polyketide synthases (PKSs) or their combinations could be involved in the biosynthesis of these phthalides[46,47]. We therefore examined the

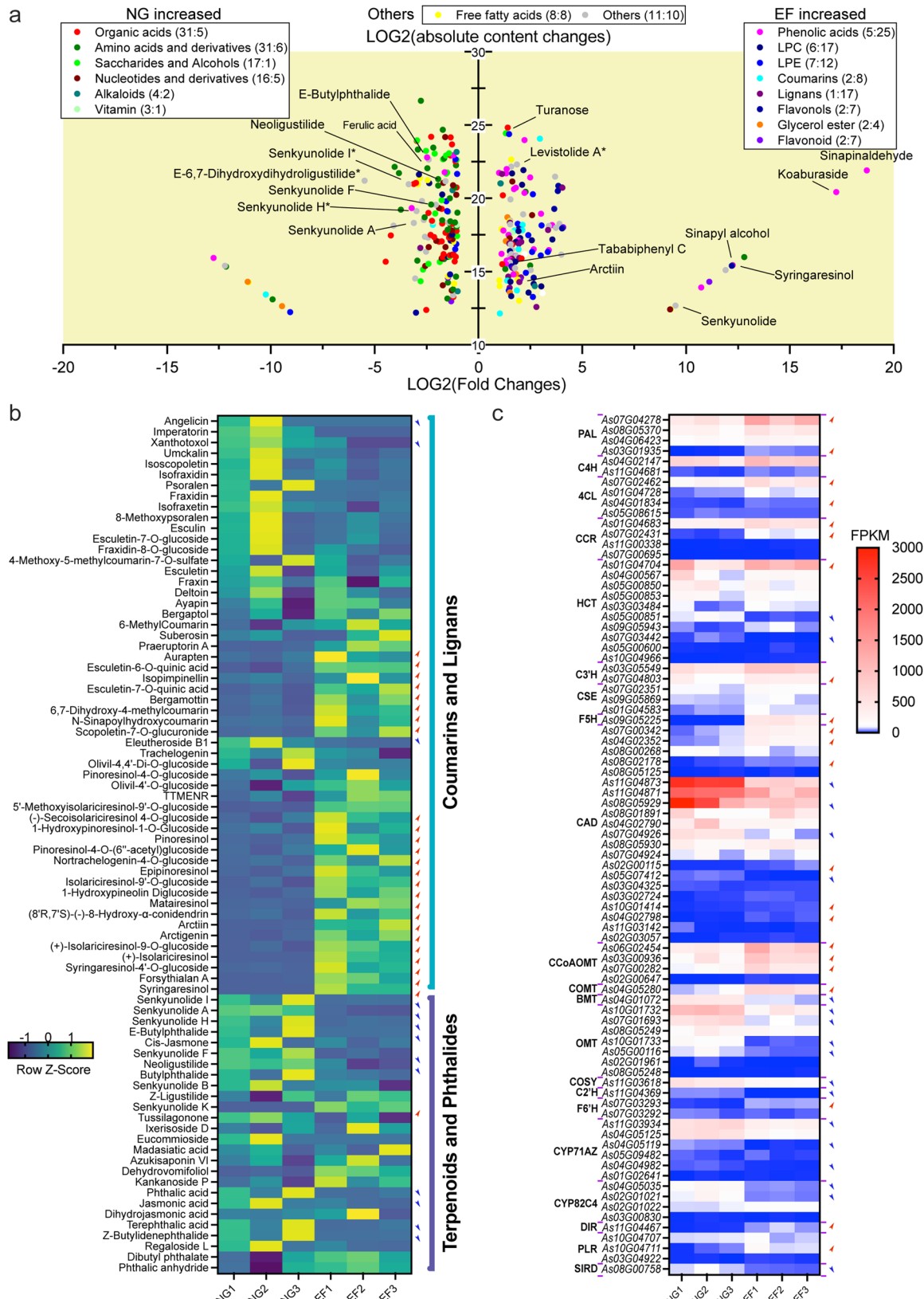

*A. sinensis* genome together with transcriptome and metabolite profiling for the biosynthesis of ligustilide and butylidenephthalide and other monoterpene volatiles that contribute to the medicinal functions of Angelica roots.

To more clearly profile bioactive components in Angelica roots, volatile terpenoids, and phthalides were examined by using headspace solid-phase microextraction-gas chromatography-mass spectrometry (SPME-GC-MS). The volatiles of early-flowering (EF) and normally growing (NG) roots showed notable differences. In addition to the higher levels (~47% of total volatiles) of Z-ligustilide and Z-butylidenephthalide and their E-type isomers as major components in NG roots, the EF roots of *A.*

**Fig. 4 Metabolic landscape of important bioactive ingredients and the related gene expression in Angelica roots. a** Changes in metabolites between NG and EF samples. The horizontal axis shows log2-fold changes, and the vertical axis shows log2 absolute content changes. The dot colors represent the different compound classes. Numbers in brackets indicate the number of compounds upregulated in NG and EF samples. **b** Heatmap of the contents of metabolites "Coumarins and lignans" and "Terpenoids and phthalides" with different contents between the NG and EF groups. The data were normalized by the Z score in rows. The red and blue arrows indicate the upregulated and downregulated metabolites, respectively (VIP ≥1 and LOG2 (fold change) ≥1 or ≤−1). **c** Heatmap showing differential gene expression related to coumarin, lignan and lignin biosynthesis between NG and EF samples in Angelica roots. The red and blue arrows indicate the upregulated and downregulated genes (LOG2 (fold change) ≥1 or ≤−1 and $p$ ≤0.05), respectively. Only the genes with FPKM ≥5 in at least one sample are shown.

*sinensis* also contained fewer phthalides (34% of total volatiles), as well as much less abundant monoterpenes, such as α-pinene and E-β-farnesene, (Figs. 6a, b). These data indicated that early bolting and flowering also negatively impacted volatile accumulation in Angelica roots.

Genome analyses revealed that three key gene families involved in the MEP pathway toward monoterpene synthesis, *MCT, HDS,* and *HDR*, were expanded in the *A. sinensis* genome in comparison with the Arabidopsis and grapevine genomes (Supplementary Data 4). *A. sinensis* genome sequences revealed an extremely enhanced monoterpene pathway during the evolution of several genera in the Apiaceae family (Supplementary Data 4), which is consistent with the diverse and enriched monoterpene volatile profiles in these plants (Fig. 6a).

Transcriptome data showed that genes involved in glycolysis and the pentose phosphate pathway were downregulated in EF Angelica roots, which also negatively affected the mavalonic pathway (MVP) and 2-C-methyl-erythrose 4-phosphate (MEP) pathway, leading to the biosynthesis of mono-and sesquiterpenoids (Fig. 6c). The *DXS, MDS, CMK,* and *HDR* genes involved in the plastic MEP pathway, one *IPPI* and two *GPPS* genes for monoterpenoid biosynthesis were significantly downregulated in EF Angelica roots compared with NG Angelica roots (Fig. 6c).

**Gene family of terpenoid synthase genes.** *A. sinensis* is a triennial medicinal plant that typically flowers in its third year but can flower early in May of its second year (Fig. 7a). As Angelica roots contain a wide range of terpenoid volatiles at abundant levels, they are also regarded as major components contributing to clinical functions[48]. Terpenoid synthase family genes play key catalytic roles in plant terpenoid biosynthesis. A total of 28 putative TPS genes in the *A. sinensis* genome belonging to five TPS subfamilies (TPS-a, TPS-b, TPS-c, TPS-e/f, and TPS-g) were identified (Fig. 7b). The TPS-b family was expanded in both *A. sinensis* (15) and *C. sativum* (20), and the expansion of TPS-b genes in the *A. sinensis* genome was mainly due to tandem duplication (Ks<0.1). There were 5 more TPS genes in *A. sinensis* (QH) than in *A. sinensis* (GS), which indicated that the completion of *A. sinensis* (QH) was better than that of *A. sinensis* (GS). We detected 8 TPS genes that were expressed in Angelica roots (FPKM ≥1 at any samples), and most of them had higher expression levels in NG roots than in EF roots (Fig. 7b).

**Polyketide synthases (PKSs) were expanded in the *A. sinensis* genome.** To further verify the possibility that PKSs are involved in the biosynthesis of the polyketide derivatives ligustilide and butylidenephthalide in *A. sinensis*, we analyzed genes that are involved in the biosynthesis of acetyl-CoA and malonyl CoA, which are used as substrates for type II and III PKSs for the production of polyketides (Fig. 8a)[46,47]. Acetyl-CoA carboxylase (ACC) is the main enzyme catalyzing the conversion of glycolysis pathway-derived acetyl-CoA into malonyl CoA, which is a key intermediate for fatty acid, polyketide, and flavonoid biosynthesis[47]. Plant ACC is composed of two subunits, the biotin

carboxylase and carboxyl transferase subunits[47]. The coding genes for two ACC subunits, BCCP2 (CAC1) (4) and CAC2-CAC3 (5), were expanded in the *A. sinensis* genome in comparison with the Arabidopsis and grapevine genomes, respectively (Table S12). Consistent with lower Z-ligustilide and Z-butylidenephthalide levels in EF Angelica roots, at least two ACC subunit genes were downregulated in EF roots compared with NG roots (Fig. 8b).

PKS consists of a large gene family encoding multifunctional enzymes that catalyze condensation of malonyl CoAs or malonyl CoA with other acyl CoAs to generate diverse polyketides[46,47]. In particular, type III PKS (TKS) catalyzes linear tetraketide-CoA synthesis with hexanoyl-CoA and malonyl CoA and might provide a backbone for Z-ligustilide and Z-butylidenephthalide biosynthesis[49]. A previous study showed that a TKS olivetolic acid cyclase (OAC) catalyzed a C2–C7 intramolecular aldol condensation with carboxylate retention in the linear tetraketide-CoA to form olivetolic acid in *Cannabis sativa*[49]. OAC was structurally similar to Z-ligustilide and Z-butylidenephthalide, with only differences in the position of the olefinic link and hydroxyl group[49]. A multifunctional protein (MFP) could handle the switch of olefinic links and hydroxyl groups in the lipid metabolism process[50]. It has thus been proposed that Z-ligustilide and Z-butylidenephthalide are synthesized via a similar mechanism through the PKS pathway, although the exact enzyme or gene responsible for their biosynthesis remains unknown. In the *A. sinensis* genome, PKSs also formed a large gene family of 120 members, among which the type III PKS genes are expanded (Table S13 and Fig. 8d).

Transcriptome analyses showed that four PKS genes, namely, *As05G08873, As11G04238, As10G03800,* and *As08G02849,* were highly expressed in Angelica roots (Fig. S4), and in particular, we also found that some of the PKS genes were repressed in EF Angelica roots as compared with NG roots (Fig. 8d), indicating that these PKSs might be involved in the biosynthesis of phthalides. The overall expression of *ACC* and *PKS* genes in Angelica roots was lower in EF plants (Fig. 8c). Further studies with isotope-labeled substrates in tracer experiments, together with enzyme and molecular approaches, are needed to unveil the mechanism underlying the biosynthesis of Z-ligustilide and Z-butylidenephthalide in *A. sinensis*.

## Discussion

Many details about biosynthesis pathways leading to the production of pharmaceutically important natural products, such as phthalides, coumarins, and terpenoids, in Angelica roots or even other species of the Apiaceae family remain unclear, and their underlying molecular mechanisms are poorly understood[1,13,51,52]. We thus assembled a high-quality chromosome-level genome for *A. sinensis* var. Qinggui1 and dissected the genetic basis of the biosynthesis of natural products and the herbal medicinal nature of Angelica roots. Comparisons with the previously published *A. sinensis* (GS) genome showed that our genome assembly and gene prediction were of higher quality and had greater representativeness, making the genome a valuable

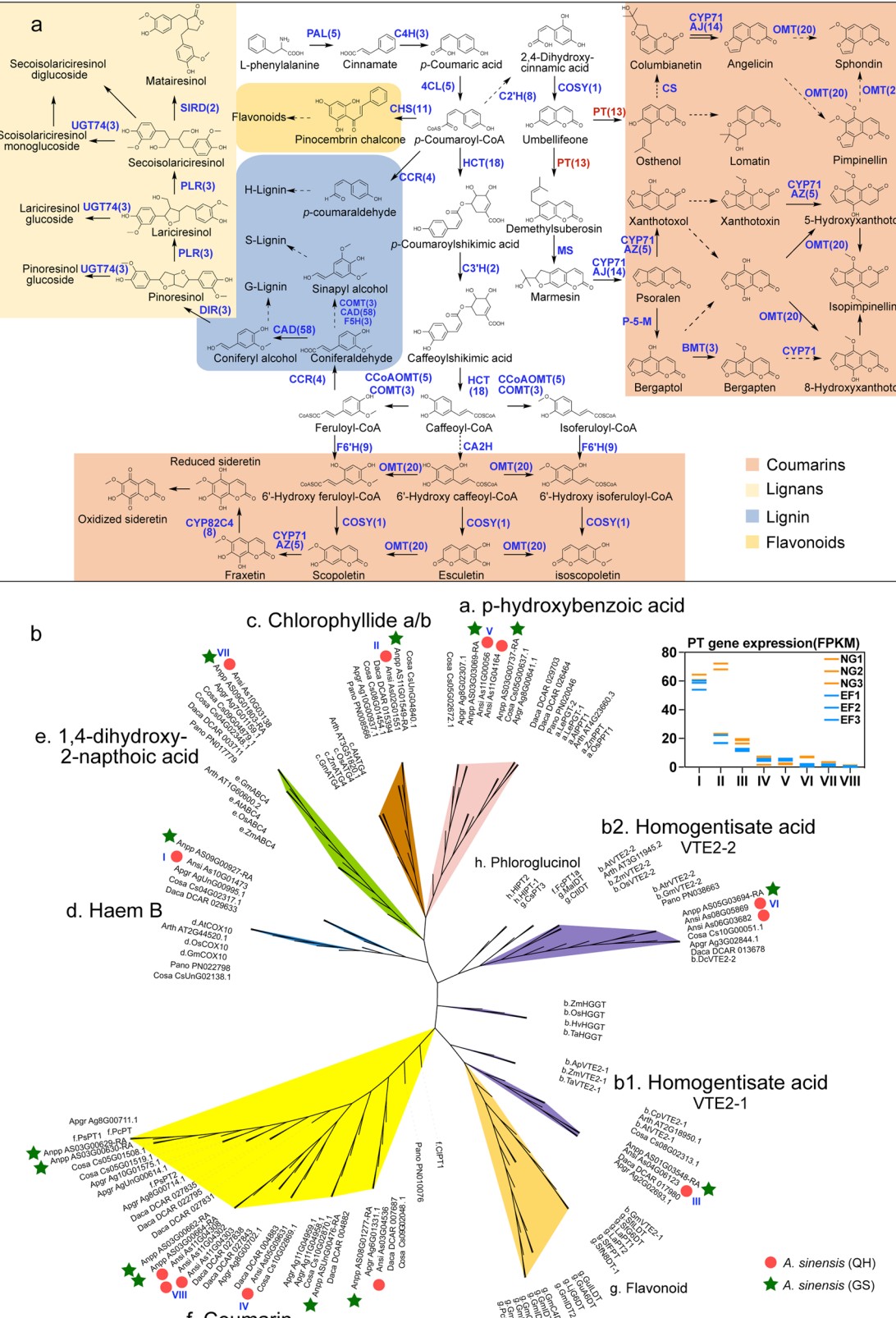

**Fig. 5 The biosynthesis pathways of phenolics and evolutionary analysis of prenyltransferase (PT) genes in *A. sinensis*. a** Putative biosynthesis pathways of coumarins, lignin, lignans and flavonoids. The numbers in parentheses indicate the number of genes. Different background colors represent the synthetic pathways of different products. The PT genes are highlighted in red. The genes in different gene families are listed in Supplementary Data 3. **b** Rootless phylogenetic tree of PT genes. The tree shows the grouping of PT genes according to the type of substrate (**a–h**). The orthologous genes in *A. sinensis* (QH) and *A. sinensis* (GS) are highlighted. The genes in the **c** and **d** subtrees had relatively high expression levels.

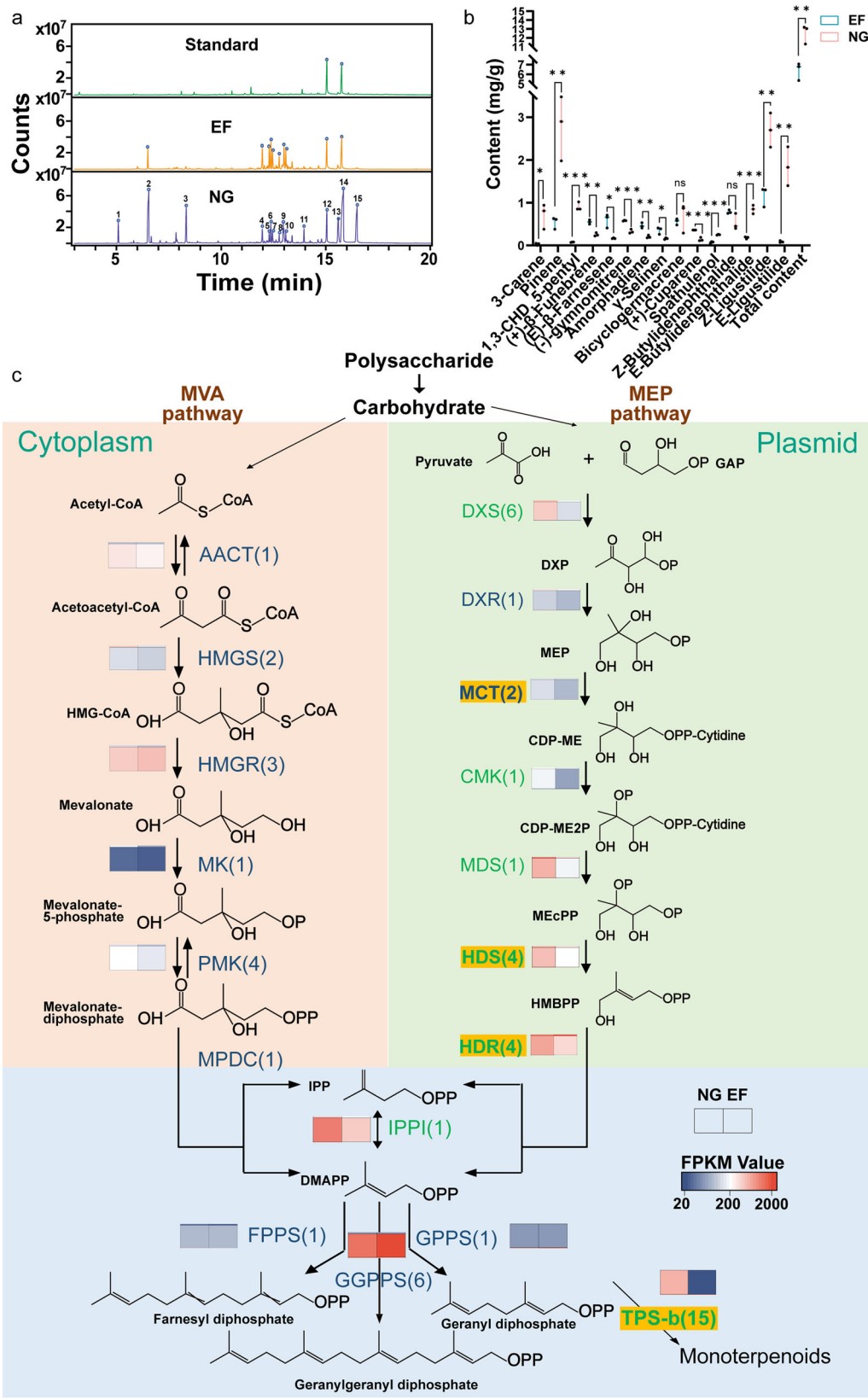

resource for future genome evolution studies and detailed functional genomics studies for the improvement of the medicinal quality and agronomic traits of *A. sinensis*.

The genome that we assembled had several advantages over the published *A. sinensis* (GS) genome. The size of our genome was closer to the size estimated by flow cytometry[13]. Comprehensive comparisons of our genome with the *A. sinensis* (GS) genome revealed high collinearity between these two genomes except for a large inversion along homologous chromosomes Chr09 (our genome) and Chr04 (*A. sinensis* (GS)). Genome collinearity between these two genomes and *A. graveolens* showed that this inverted region was well aligned between our genome and *A.*

**Fig. 6 Biosynthesis and regulation of phthalides and other terpenoid volatiles in Angelica roots. a** Headspace solid-phase microextraction-gas chromatography-mass spectrometry (SPME-GC-MS) analysis of the contents and composition of volatiles in Angelica roots from early-flowering (EF) and normally growing (NG) plants. **b** Differential content analysis of the volatiles in Angelica roots between EF and NG plants. **c** Enzymatic reactions in the mevalonate (MVA) and methylerythritol phosphate (MEP) pathways in plants and synthesis of short-chain prenyl diphosphates. The MVA pathway is shown in light red; the MEP pathway is shown in light green. Abbreviations and full names are given in Table S16. Data are expressed as the means ± SDs from at least three independent experiments with triplicates. Differences between NG and EF samples are considered significant when $**P < 0.01$ and $*P < 0.05$ in Student's $t$ test.

graveolens but not between _A. sinensis_ (GS) and _A. graveolens_, suggesting that either _A. sinensis_ (GS) had an assembly error in this region or that this was a unique feature of the _A. sinensis_ (GS) genome. Although we annotated fewer high-confidence genes, we used the same BUSCO version and parameters to compare the completeness of the gene sets and found that our genome had 96.41% complete genes, while _A. sinensis_ (GS) had only 88.10%, which indicated that our gene set might be more reliable. Finally, subsequent analyses of multiple gene families also confirmed that our gene set was more complete. For example, we identified 5 more PT genes (Fig. 7), 3 more C2′H genes, 6 more F6′H genes (Fig. S5), and 9 more CYP71AJ genes (Fig. S6).

With the published _A. sinensis_ (GS) genome, the authors reported that _A. sinensis_ was more ancestral and diverged earlier than _C. sativum_ and _A. graveolens_[13]. We provide several lines of evidence to challenge this inferred evolutionary position. First, we started with 20 representative plant species from 14 families and 12 orders. We selected 2,133 one-to-one orthologous gene families that were shared by all the species to construct a phylogenetic tree. This tree shows that within the Apiaceae family, carrot is the most basal, followed by celery, then coriander, and finally Angelica. Second, from the analysis of 4DTV and KS, we found that for all WGD events, WGD occurred earlier in carrot than in celery, coriander, and then in Angelica. WGDs are a major driving force for species diversification[31]. This evidence also supports the new evolutionary relationships within the Apiaceae family. Finally, we clustered the genes of _A. sinensis_ (QH), _A. sinensis_ (GS), coriander, celery, carrot, and _P. notoginseng_ into gene families. We obtained 3,189 single-copy genes, with which to construct a phylogenetic tree and estimate the divergence time. This phylogenetic tree and species divergence time were consistent with the previous phylogenetic tree based on 20 angiosperm plants and further contradicted the evolutionary relationships in the _A. sinensis_ (GS) article. This improves our understanding of the evolutionary relationships of different species in Apiaceae.

Previous transcriptomic and metabolomic studies of early bolting and flowering in _Angelica sinensis_ have identified a number of differentially expressed genes and metabolites[15,18,53,54]. For example, Li et al.[18] identified over 2,600 differentially expressed genes (DEGs) between bolted and unbolted plants, and found that genes associated with floral development and sucrose metabolism were coordinated to regulate flowering. However, these studies have been limited by the use of low-throughput methods and the lack of a complete genome assembly. Based on this improved Angelica genome assembly, together with targeted and nontargeted metabolomics, transcriptome analyses of Angelica roots under two statuses, NG and EF, showed drastic differences in the biosynthesis and accumulation of pharmaceutically important secondary metabolites, including phthalides, coumarins, lignans, and terpenoids, we were able to dissect their biosynthesis pathways and discover the putative key genes in Angelica roots. Genomic expansions, differential expression patterns, and evolutionary tracks of key metabolic genes, such as PTs, OMTs, UGTs, and CYP71s for coumarin and lignan biosynthesis, TPSb for terpenoid

biosynthesis, and PKSs for polyketide biosynthesis, in comparison with their counterparts in other species of the Apiaceae family, seemed to explain very well the metabolomics data on the traditionally regarded different medicinal values and qualities of EF and NG Angelica roots.

As various volatiles in Angelica roots contribute to the major bioactivities of the herbal medicine, we explored the genomic basis of the biosynthesis of these chemically enriched and structurally diverse volatiles. The plastidic MEP pathway leading to monoterpene biosynthesis was specifically evolutionarily enhanced in _A. sinensis_, since several key metabolic genes, such as the _MCT_, _HDS_, _HDR_, and _TPS-b_ gene families, were expanded in the _A. sinensis_ genome. Consistent with the reduced quality of EF Angelica roots, we found that both the contents of FA and monoterpenes were significantly decreased, and their biosynthesis genes were downregulated compared with those in regular roots. Two major bioactive volatiles, Z-ligustilide and Z-butylidenephthalide, were also reduced in EF Angelica roots, consistent with the repressed expression of two key polyketide biosynthesis genes encoding ACC subunits and PKSs. Although the Z-ligustilide and Z-butylidenephthalide biosynthesis pathways remain unsolved, our study showed that ACCs that could catalyze malonyl CoA biosynthesis and PKSs that condense malonyl CoAs, similar to other long-chain acyl CoAs, might be the key enzymes involved in the biosynthesis of the two phthalides. Z-Ligustilide and Z-butylidenephthalide are the major contributors to the medicinal functions of _A. sinensis_ herbs. This insight could facilitate future studies with isotope-labeled tracer experiments and functional characterization of these key enzymes/genes.

## Methods

**Plant preparation and DNA sequencing.** In this study, we used the _Angelica sinensis_ cultivar Qinggui 1 for genome sequencing and assembly. This cultivar was an important cultivar that has been planted for hundreds of years in Qinghai province, China. Seeds were collected from the Qinghai Key Laboratory of Crop Molecular Breeding at the Northwest Institute of Plateau Biology of the Chinese Academy of Science. A. sinensis is a triennial medicinal plant that typically flowers in its third year but can flower early in May of its second year. Seeds were sown in July and August and sampled in April and May of the following year at Diyao village in Dayuan Town, Huangzhong County, Xining City, Qinghai Province (~2700 m above sea level; 36°13′ 32″ ~ 37°03′19″N and 101°09′32″ ~ 101°54′50″E). Plants were grown under similar field conditions and those exhibiting an early flowering phenotype were selected as EF samples while those in normal vegetative growth were used as NG samples.

To improve the accuracy of gene prediction, transcriptome sequences were generated from four tissues of the _A. sinensis_ var. Qinggui1: root, stem, leaf and seed. To study the dynamic changes in the transcriptome between two developmental statuses - normal vegetative growth (NG) and early flowering (EF) - 21 samples were used for RNA sequencing. These included three replicates each of root, leaf and stem tissue at both NG and EF statuses as well as three replicates of flower tissue at the EF status.

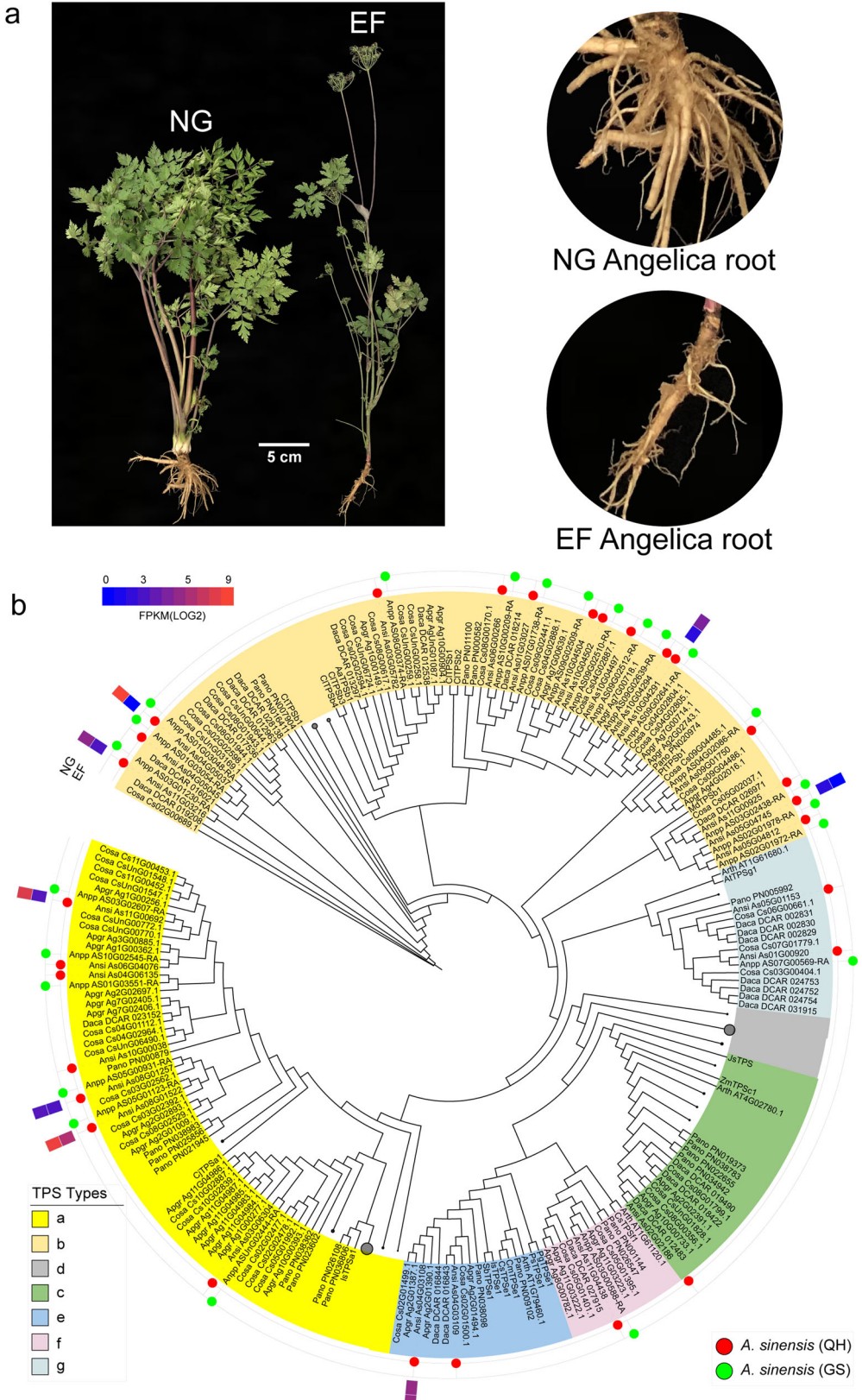

**Fig. 7 The different growth statuses of Angelica plants and phylogenetic tree of TPS genes. a** Plants were sown simultaneously and grown in the same environment. Samples were taken at the same time for observation and analysis. EF early flowering, NG normal growth. We highlight the highly lignified Angelica root of the EF plant and the normally developed storage root of the NG plant on the right side. **b** Five TPS subfamilies (TPS-a, TPS-b, TPS-c, TPS-e/f, and TPS-g) were clearly identified. The genes from *A. sinensis* (QH) and *A. sinensis* (GS) are highlighted by red and green dots, respectively. The heatmap of gene expression is illustrated.

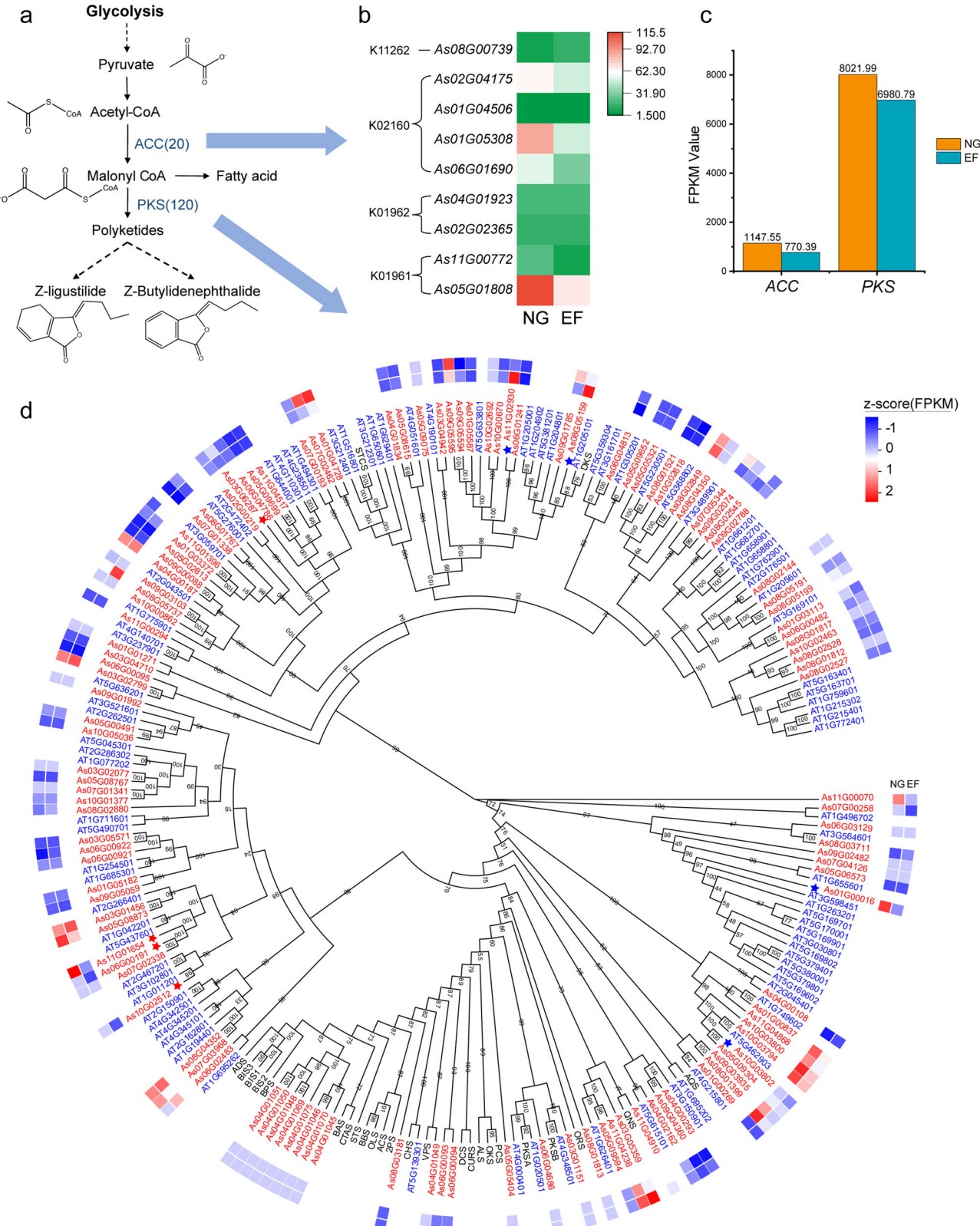

**Fig. 8 Biosynthesis and regulation of phthalides. a** The malonyl-CoA biosynthesis metabolic pathway. **b** Heatmap displaying the expression of typical *ACC* genes in Angelica roots between EF and NG plants. **c** The overall expression (FPKM) of ACC and PKS genes in Angelica roots between EF and NG plants. **d** Phylogeny of polyketide synthase genes (PKSs). The heatmap displays the gene expression in Angelica roots between EF and NG plants. The color of gene IDs shows the source of different species: red: *A. sinensis*; blue: *A. thaliana*; black: seed sequences. The red stars highlight the upregulated genes, and the blue stars highlight the downregulated genes.

Additionally, six root samples (three replicates each at NG and EF statuses) were used for metabolome identification. All samples were carefully collected and rapidly frozen in liquid nitrogen and stored at −80℃.

High molecular weight (HMW) genomic DNA was isolated from fresh Angelica sinensis leaves using a DNeasy Plant Mini Kit (Qiagen, United States). More than 50 μg of this DNA was used to generate SMRTbellTM libraries for long-read sequencing on the PacBio Sequel II platform using the "circular consensus sequencing (CCS)" mode.

Hi-C libraries were also constructed from fresh A. sinensis leaves following a previously published procedure 1. Plant tissues were fixed with formaldehyde and cross-linked DNA was digested overnight with DpnII. The sticky ends of the resulting fragments were biotinylated and ligated randomly to create chimeric fragments representing the original cross-linked physical interactions. These fragments were then processed into paired-end sequencing libraries and sequenced on an Illumina Hiseq 4000 platform with 2 ×150 bp reads.

Total RNA was extracted from leaves, roots, stems, seeds, and flowers using the Trizol extraction method (Invitrogen, CA, USA) and purified with magnetic beads (Invitrogen, CA, USA). The purified mRNA was fragmented and used as a template to synthesize first-strand cDNA with random primers. Second-strand cDNA was then synthesized by adding DNA polymerase I, RNase H, dNTPs and buffer. The resulting cDNA fragments were purified with a QiaQuick PCR extraction kit and subjected to end repair and poly(A) tail addition before being ligated to Illumina sequencing adapters. Suitable fragments were selected by size and used as templates for PCR amplification to complete the cDNA library construction. Transcriptome sequencing was performed on an Illumina HiSeq 4000 platform.

**Genome size estimation**. To evaluate the genome size of A. sinensis, an Illumina high-throughput sequencing library with an insert size of 350 bp was constructed from genomic DNA and paired-end sequenced on the Illumina HiSeq 4000 platform. Raw reads were preprocessed to obtain clean data using SOAPnuke[55] (version 1.5.6). Kmer analysis method was used for A. sinensis genome size estimation. Kmerfreq from GCE v1.0.2[56] were adopted to obtain the 17-mer distribution from clean NGS reads with the parameter of -k 17 -L300. The output file was calculated and visualized using line_diagram.pl in kmerfreq package to estimate the genome size and heterozygosity rate.

**Genome assembly and evaluation**. The draft genome was assembled using the raw reads generated by the PacBio and Illumina sequencing platform. First, PacBio long reads were corrected and the genome framework was assembled using the NextDenovo (version 2.3.0) with default parameters. Primary contigs were polished with illumine short reads using NextPolish[57] (version 1.3.0) to correct any remaining errors, yielding a final draft genome assembly. To generate a chromosome-level genome assembly, Hi-C technology was used to anchor the contigs onto chromosomes. First, the high-quality paired-end Hi-C sequencing reads were mapped to A. sinensis draft genome by BWA[58] (version 0.7.17). Then, HiC-Pro[59] was used for repetitive reading removal, classification, and quality assessment. Raw counts of Hi-C links were aggregated and separately by using Juicer[60] and 3D-DNA[61]. Finally, based on the threshold of contact frequency, the contigs were clustered into 11 pseudo-chromosome linkage groups and the results were carefully manual checked using the Juicebox[62] (version 1.11.08).

To evaluate the consistency and nucleotide accuracy of the genome assembly, The short reads were mapped to the assembled genome using BWA[58] software. The SNPs and indels were identified by ReSeqTools[63] and GATK[64] (version 3.3.0) followed the best practice workflow of GATK and filtered using VCFtools[65] with parameters "--min-meanDP 2 --max-meanDP 520 --min-alleles 2 --max-alleles 2". The integrity of the genome assembly was evaluated by Benchmarking Universal Single-Copy Orthologs (BUSCO, version 5.0)[20].

**Genome annotation**
*Transposable element annotation*. Repetitive elements in the A. sinensis genome were identified using both homology alignment and ab initio prediction methods. De novo libraries of repetitive elements were constructed using LTR_Finder[66], PILER[67], and RepeatScout[68]. These libraries, combined with the Repbase[69] database, were used to search for repeat sequences in the genome using RepeatMasker[70]. Tandem repeats were predicted using Tandem Repeat Finder (TRF).

*Protein-coding gene prediction and gene functional annotation*. Three methods were combined for protein-coding gene prediction in A. sinensis: homology-based, transcriptome-assisted, and ab initio prediction. For homology-based prediction, annotated protein sequences from *Arabidopsis thaliana*, *Coriandrum sativum*, *Apium graveolens*, *Daucus carota*, *Glycine max*, *Helianthus annuus*, *Lactuca sativa*, *Oryza sativa*, *Solanum lycopersicum*, and *Vitis vinifera* were mapped to the A. sinensis genome using TBLASTN (v2.2.29+) with an E-value cutoff of 1e−5. BLAST hits were concatenated using Solar software (v0.9.6) and genomic regions corresponding to 1000 bp upstream and downstream of each candidate gene were extracted for gene structure prediction using Genewise[71] (v2.4.1). For transcriptome-assisted prediction, RNA-seq data from 25 samples were mapped to the assembled genome using HISAT2[72] (version 2.21) and potential transcripts were assembled using StringTie[73] (version 1.3.3b). Transcriptome proteins were predicted using Transdecoder (version 5.5.0). For ab initio prediction, AUGUSTUS[74] (version 3.2.1), SNAP[75], and glimmerHMM[76] were used on the repeat-masked genome with specific parameters obtained by training with homology-based gene sets. All gene models were integrated into a consensus set using EvidenceModeler[77] (version 1.1.1) with weights assigned to each type of evidence. Functions of predicted proteins were obtained by BLAST against InterproScan[33], Gene Ontology (GO)[34], Kyoto Encyclopedia of Genes and Genomes (KEGG)[35], SwissProt[36], TrEMBL, KOG, and non-redundant protein NCBI databases.

*Non-coding RNAs*. Canonical small non-coding RNAs, including ribosomal RNAs (rRNAs), transfer RNAs (tRNAs), small nuclear RNAs (snRNAs), and small nucleolar RNAs (snoRNAs), were identified in the A. sinensis genome using BLAST against various RNA libraries. tRNAs were predicted using tRNA scan-SE[78] (version 1.3.1) while snRNAs and microRNAs (miRNAs) were identified by alignment with BLAST followed by searching with INFERNAL[79] (version 1.1.2) against the the Rfam database[80] (v12.0).

**Distinguish of high confidence and low confidence genes**. The high confidence (HC) gene set was supported by homology with gene sets from Lactuca sativa, Apium graveolens, Daucus carota, Helianthus annuus, Solanum lycopersicum, Glycine max, Arabidopsis thaliana, Vitis vinifera, and Oryza sativa (identity ≥40% and coverage ≥50%) or by transcriptome data (FPKM ≥ 1 in 25 sequenced RNA samples). HC genes had complete open reading frames (ORFs) and lacked transposable element (TE) sequences. In contrast, the low confidence (LC) gene set was not

supported by homology or transcriptome data and may have contained TEs. A total of 41,040 genes were identified as HC while 27,767 were identified as LC. BUSCO analysis showed that the completeness of the whole gene set was 96.41%, with the HC gene set at 95.71% and the LC gene set at 0.81%, indicating the high quality of the HC gene set.

**Transcriptome sequencing and gene differential expression analysis.** RNeasy Mini Kits (QIAGEN Sciences Inc., USA) were used to isolate total RNA from the tissues according to the manufacturer's instructions. The main steps were as follows: (a) total RNA sample extraction; (b) library construction; and (c) sequencing and bioinformatics. The raw reads were filtered by SOAPnuke[55] (version 1.5.6) and then aligned to the genome using Bowtie2[81]. Gene expression was quantified using RESM[82] and gene differential expression analysis was conducted using the DESeq2 package[83]. The functional enrichment analyses of Gene Ontology and KEGG pathway data were carried out by in-house Perl scripts.

**Gene families.** The protein sequences of *A. sinensis* (QH) and other six plant representative species, including *A. sinensis* (GS), *A. graveolens, C. sativum, D. carota, Panax notoginseng,* and *Arabidopsis thaliana* were selected for gene family analysis. The longest protein sequence was employed if a gene had multiple alternative splicing transcripts, and short protein sequences (<50 bp) were also discarded. Then all-against-all comparisons were performed using BLASTP (E value cutoff: $1e^{-5}$), and OrthoFinder[37] was used to cluster the BLASTP results to obtain single-copy and multi-copy gene families. The 4DTv mutation rate was calculated for each of the homologous gene pairs between and within species. Ks density of paralogous and orthologous gene pairs was calculated using WGDI[84]. To analyze the evolution of gene families, CAFE[85] as used to calculate the number of expansions and contractions of gene families. In addition, the gene collinearity within the *A. sinensis* genome was detected using MCScan X[86]. Transcription factor (TF) genes were identified by iTAK[21] pipeline.

**Phylogenetic tree analysis of 20 species**
*Ortholog identification.* To obtain alignments of orthologous genes across all species, a clustering and phylogenetic-based orthology inference method[87] was used. We performed an all-by-all BLASTP with an e-value of $1e^{-5}$ using the proteins of all individuals as both query and subject. Up to 1000 best hits for each sequence were retained, and blast hits where the length of the alignment was smaller than half of the length of either the query or the subject sequences were excluded to remove hits to conserved motifs and short sequence fragments. Blast hits were clustered with MCL with an inflation value of 1.4[88]. Clusters with less than 12 tips were discarded, and the remaining clusters were aligned with MAFFT v7.310 using the accuracy-oriented method E-INS-i and a maximum of 1000 iterative refinement cycles[89]. Sequence alignments were trimmed with PHYUTILITY v.2.2.6 to remove positions with more than 10% missing data[90]. Cleaned alignments were used in RAXML v8.2.11 to estimate a phylogenetic tree with the PROTCATWAG model of amino acid substitution[91]. Branches greater than 0.6 were cut, and spurious terminal branches greater than 0.2 and more than ten times longer than their sisters were also removed. Monophyletic tips were masked, keeping the sequence with the shortest terminal branch length. Internal branches longer than 0.3 expected substitution per site were cut, and only subtrees with 12 or more species kept. Finally, the homologous gene trees were pruned to obtain one-to-one orthologous gene families (one sequence per

species for each orthologous gene family) using the Monophyletic outgroups (MO) method[87].

*Phylogenetic inference.* To construct the phylogenetic relationships of 20 species, two alternative approaches were employed: ML phylogenetic inference on the concatenated data set of all orthologs (supermatrix approach) and coalescent-based phylogenetic reconstruction using individual gene-trees (species tree approach). Only orthologous genes available in all 20 species were used for all phylogenetic analyses. For each orthologous gene, they were re-aligned with PRANK v.170427 using the protein substitution matrix method[92], and alignments were trimmed with PHYUTILITY v.2.2.6 to remove positions with more than 70% missing data. For the Maximum-Likelihood (ML) phylogenetic reconstruction, the alignments of all orthologous genes were concatenated, and each gene was treated as a separate partition. The ML tree was estimated using IQ-TREE v.1.5.5[93] under the WAG model, and supports were evaluated with ultrafast bootstrapping testing (1000 replicates)[94]. Gene trees support ratios for this ML tree were obtained using ASTRAL v.5.5.9 with parameters setting: -t 1[95]. For the coalescent approach, single ML gene trees were inferred using RAxML with the PROTCATWAG model. The species tree was inferred using ASTRAL with parameters setting: -t 1 –gene-only, and branch lengths for this tree were obtained using IQ-TREE with the same settings as with the "supermatrix" approach.

*Divergence time estimation.* Divergence times for each orthologous gene were estimated using Bayesian inference in MCMCTree v.4.9j package[96] with the uncorrelated relaxed clock model (clock = 2). The ML estimates of the branch lengths were calculated using BASEML using the Empirical+F protein substitution model. The prior on the overall substitution rate across loci (μ) was set to G (1, 20), and the prior for the degree of rate variation across branches ($\sigma^2$) was set to G (1, 4.5). The time unit was set to 100 Ma, and the parameters for the birth-death process were set as $\lambda = \mu = 1$ and $r = 0$. The MCMC analyses were run for 1 million generations sampled every 100 generations after a burn-in of 50 000 iterations. The chain convergence was assessed by running MCMC analyses twice, and the effective sample size of all parameters was confirmed to be >200 using Tracer v.1.7 (http://tree.bio.ed.ac.uk/software/tracer/). Three fossil calibration points were used: the divergence time of *V. vinifera* and *A. thaliana* is 107 million years ago to 135 million years ago, the divergence time of *S. lycopersicum* and *A. thaliana* is 111 million years ago to 131 million years ago, the divergence time of *S. lycopersicum* and *D. carota* is 95 million years ago to 106 million years ago. Finally, all orthologous gene trees were integrated using DensiTree.v2.2.7 software[97].

**Whole genome comparisons and gene family analysis.** Whole-genome comparisons, visualization of the alignments and genetic variation identification were performed using MUMmer[98] (version 4.0). Protein sequences from 6 species were selected for gene family analysis: *P. notoginseng, D. carota, A. graveolens, C. sativum, A. sinensis* (GS), and *A. sinensis* (QH). We used *P. notoginseng* as the outgroup and clustered the genes with OrthoFinder[37] v2.5 (default parameters). We extracted 3189 single-copy genes and used them to construct a phylogenetic tree. The concatenated amino acid sequences were aligned using Muscle (v 3.8.31), and the conserved sites were extracted from the multiple sequence alignment results using Gblocks (Version 0.91b) (Parameters: -b4 = 5  -b5=h  .t = p  .e = .Gblocks). The sequences were sorted and merged using seqkit (v 2.1.0). The merged fasta file was converted into phy format, and the optimal

amino acid substitution model was determined using prottest (v 3.4.2) (parameters: -f a -N 1000 -all-distributions -F -AIC -BIC -tc 0.5). Based on the predicted optimal model, we constructed the tree using RAxML (v 8.2.11) (parameters: -N 1000 -m PROTGAMMAIJTTF -k -O -o Pano). The 4DTv mutation rate was calculated for each of the homologous gene pairs between and within species. The Ks density of paralogous and orthologous gene pairs was calculated using WGDI[84]. To analyze the evolution of gene families, CAFE[85] was used to calculate the number of expansions and contractions of gene families. Gene collinearity was detected using MCScanX[86] and visualized using NGenome-Syn v1.41[99]. Transcription factor (TF) genes were identified by the iTAK[21] pipeline.

**Identification of genes involved in coumarin, lignan, phthalide, and terpenoid biosynthesis**. The seed sequences used for orthologous gene identification were collected and are listed in Supplementary Data 5 and Tables S14 and S15. The protein sequences of *A. thaliana*, *P. notoginseng*, carrot, celery, coriander, *A. sinensis* (GS), and *A. sinensis* (QH) were used to perform BLASTP v2.11.6 (E-value < 1e-50) analyses to identify orthologous genes. Multiple sequence alignment was performed using muscle[100], and pruning was performed using trimal[101] (-gt 0.8). Tree building was performed using iqtree[102] (–mset JTT -g_stop_i 1). The tree was drawn using iTOL[103], and the topological structure of the tree was used to identify the correct orthogonal gene members. The phylogenetic trees of the important gene families in the typical Apiaceae species are listed in Figs. S5–S14. For genes with clear KEGG annotations and KO numbers, their homologous genes were determined based on an E-value of 0.0.

**Profiling of metabolites with nontarget metabolomics assay approaches**. Root samples were collected, cut into small pieces, ground into a fine powder in liquid nitrogen and mixed thoroughly. Approximately 3 g of each sample was then freeze-dried and crushed using a mixer mill (MM 400, Retsch, Haan, Germany) with a zirconia bead for 1.5 minutes at 30 Hz. Next, 100 mg of each powdered sample was dissolved in 1.0 mL of 70% aqueous methanol and extracted overnight at 4 °C on a rotating wheel. After extraction, the mixtures were centrifuged at 10,000 × g for 10 minutes, and the supernatants were isolated using a CNWBOND Carbon-GCB SPE Cartridge (250 mg, 3 mL; ANPEL, Shanghai, China) and filtered through a SCAA-104 filter (0.22 μm pore size; ANPEL, Shanghai, China) before LC-MS analysis. The LC-ESI-MS/MS system used for the analysis of Peel extracts consisted of an HPLC (Shim-pack UFLC SHIMADZU CBM30A system; Shimadzu, Kyoto, Japan) and an MS (Applied Biosystems 6500 Q TRAP; Foster City, CA, USA) equipped with a C18 column (Agilent SB-C18; 1.8 μm, 2.1 mm × 100 mm). LIT and triple quadrupole (QQQ) scans were acquired on a triple quadrupole-linear ion trap mass spectrometer (Q TRAP), AB4500 Q TRAP UPLC/MS/MS System, equipped with an ESI Turbo Ion-Spray interface, operating in positive and negative ion modes and controlled by Analyst 1.6.3 software (AB Sciex). The self-built database MWDB (Metware Biotechnology Co., Ltd., Wuhan, China) and the public database of metabolite information, as well as second-order spectra, were used for the qualitative analysis of the primary and secondary mass spectrometry data. Metabolite quantification was achieved using data from QQQ mass spectrometry in multiple reaction monitoring (MRM) mode. Raw mass spectrometry data was processed using Analyst 1.6.3 software (AB Sciex, Framingham, MA, USA) and the IntelliQuan algorithm[104]. The function of identified metabolites was annotated using the KEGG database[35]. VIP (Variable Importance in Projection) values were calculated using R package MetaboAnalystR[105]. Significantly regulated metabolites between different groups were determined by the parameters of VIP ≥1 and Log2FC (fold change) ≥1. Pearson correlation coefficient was calculated by the built-in cor function of R software and used to evaluate the correlation of biological repetitions.

**Analysis of volatiles by headspace solid-phase microextraction GC-MS**. The method used for the analysis of volatiles was previously described by Li et al.[106], where Z-ligustilide and Z-butylenephthalide were used as standards. In this study, 0.2 g of Angelica roots was mixed with 5 ml of boiling water in sealed headspace vials and then kept in a water bath kettle at 50 °C. After being in equilibrium for 10 min, an SPME fiber was exposed to the sample headspace for 40 min. The volatile compounds were desorbed at the GC-MS injector for 5 min at 230 °C. Then, GCMS-QP2010S analysis was performed to detect the compounds, and ethyl caprate (0.2 μg/g, Sigma, USA) was used as the internal standard to normalize the contents of the compounds.

**Statistics and reproducibility**. The data were obtained using three biological and technical replicates. The mean ± standard error of the mean (S.E.M.) was used to express all data unless otherwise indicated. Differential content analysis of the volatiles in Angelica roots between early-flowering (EF) and normally growing (NG) plants were compared by Student's $t$ tests. A probability level of $p$ value < 0.05 was considered statistically significant.

Differential expression analysis between two conditions/ groups was performed using the DESeq R package (1.18.0). The DESeq package provides statistical routines for determining the differential expression of digital gene expression data using a model based on the negative binomial distribution. The resulting $p$ values were adjusted using Benjamini and Hochberg's approach for controlling the false discovery rate[107]. Genes with an adjusted $p$ value < 0.05 as detected by DESeq were considered differentially expressed. Three biological replicates per condition were used.

Gene ontology (GO) analysis was performed to determine the significance of enrichment by DEGs with false discovery rate (FDR)-adjusted p values less than 0.05. Kyoto Encyclopedia of Genes and Genomes (KEGG) pathways were tested using KOBAS software to determine significant enrichment of DEGs. Terms with an adjusted $p$ value < 0.05 were considered enriched.

## Data availability
The supporting data for this study, along with the genome assembly and annotations, have been deposited in the CNGB Sequence Archive (CNSA)[108] of the China National GeneBank DataBase (CNGBdb) under accession number CNP0003198. The raw sequence data have also been deposited in the Genome Sequence Archive[109] in National Genomics Data Center, China National Center for Bioinformation / Beijing Institute of Genomics, Chinese Academy of Sciences (GSA: CRA013096). These data are publicly accessible at the respective databases. All other data are available from the corresponding author on reasonable request. The source data for graphs and charts in the main figures can be found in the Supplementary Data 6 file.

## Code availability
The source code used to carry out the functional enrichment analyses of Gene Ontology and KEGG pathway can be found in Source Code zip file.

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

## Acknowledgements

The work was financially supported by the QingHai Science and Technology Department (2020-SF-132), Science, Technology and Innovation Commission of Shenzhen Municipality of China under grant NO. JCYJ20180507183534578 and special funds for Science, Technology, Innovation and Industrial Development of Shenzhen Dapeng New District (Grant No. PT202101-28). The funding agencies played no role in the design of the study and collection, analysis, and interpretation of data and in writing the manuscript.

## Author contributions

B.L. and S.Z. conceived and designed the project. D.C., Q.Z., S.W., Y.Z., F.Z., M.Z., G.L., and Y.L. performed the experiments, S.L., X.J., M.X., C.L., K.Y., M.B., C.Z., M.Z., and J.Z. analyzed the data and interpreted the results. S.L., S.Z., and T.-Y.C. wrote and revised the manuscript. J.Z., B.L., and H.T.S. revised the manuscript. All the authors have read and approved the final manuscript.

## Competing interests

The authors declare no competing interests.
