## [Peer Review File · Communications Biology]

Reviewers' comments:

Reviewer #1 (Remarks to the Author):

The article "The genome of *Angelica sinensis* provides new insights into the evolution and production of medicinal compounds" by Li et al., presents an improved genome for *Angelica sinensis* var. Qinggui1 (*Angelica* roots), a Chinese traditional medicinal herb. The researchers decoded the genome, identified pathways and genes responsible for producing bioactive components in *Angelica* roots, and employed a multi-omics approach including metabolomic profiling and transcriptome analysis. The study revealed expansions of key genes involved in the biosynthesis of bioactive compounds, offering valuable insights for future synthetic biology and herb breeding research.

Before considering publication, it is crucial to address the numerous questions raised in the manuscript regarding the experiments and analysis. One important aspect to consider is clearly describing the differences between the previous analysis/publication and the current study, along with providing a rationale for the changes made. Additionally, a more detailed comparison is needed instead of solely focusing on the numerical differences.

For instance, the authors have addressed a critical limitation in a previous publication by utilizing a more comprehensive approach that involved a larger number of genes. While they emphasize the advantage of comparing over 2,000 one-to-one orthologous genes in their study (Line 198-211), it remains uncertain whether this increased gene count directly translates to greater accuracy in their results. It would be valuable if the authors could provide additional criteria or evidence supporting the claim that their approach yields more accurate results compared to the previously published article, beyond the mere increase in the number of genes compared.

Additionally, enhancing the manuscript can be accomplished by addressing concerns such as providing a more comprehensive explanation of the methodology utilized and presenting a clearer justification for the research undertaken.

Further comments are listed below.

Subtitle

It would be necessary to include clear and informative subtitles throughout the paper, particularly in parts where the current subtitle lacks clarity. This will help readers grasp the main points of each paragraph more effectively. e.g. In line 195, 196 "Inconsistent phylogenetic tree and differentiation time estimation in the published *A. sinensis* reference genome". I firmly believe that a more appropriate title can be selected for this section.

Phylogenetic analysis

The manuscript discusses the selection of 20 representative angiosperms for phylogenetic analysis. However, the criteria for choosing these specific species are not adequately explained. It will enhance the manuscript if the author provides more detailed information regarding the 'selection process'. Given the abundance of important medicinal species, it is important to clarify why certain species were chosen and whether other significant

Plant preparation and DNA sequencing

I suggest the author provide clearer details regarding the 'age' and 'timing of plant collection' for sequencing. The phrase "same place, same time, right time" is vague and could be improved by specifying the specific age of the plants at the time of sample collection. Additionally, it would be necessary for author to include information about the growth conditions such as the light cycle,

humidity, and age of the plants at the time of sample collection.

Citations

The author should consider including appropriate citations where necessary to support their statements. For instance, in line 254, when referring to "Han's article", author should cite Han's article in this section. It is one example, and I found several places author need to include appropriate citations.

Line 264

It would be helpful to specify which common databases are used for this particular part of the study.

Line 637, 646

The author introduced key metabolite genes, and I believe that it would be necessary for authors to provide more descriptive information to aid the understanding of readers who are unfamiliar with the metabolic pathways. Consider including figures or tables to visually illustrate the pathways and provide further context, instead of presenting a list without additional explanation. This would enhance the clarity and accessibility of this section

Line 745

There is more than one space between S18 and S19.

Reviewer #2 (Remarks to the Author):

This study generated the chromosome scale genome of *A. sinensis*. The study also provides a large set of omics data for root in *A. sinensis*. Since a genome sequence manuscript has already been published in "the Plant Journal", in this study, the authors have performed comprehensive comparisons with the published data and claimed it is an improved chromosome-level genome for *A. sinensis*. However, the data presentation still needs lots of improvement to fully support the conclusion. The following suggestions may help to improve the manuscript, which is encouraged to resubmit with revisions. The major difference of the two manuscripts is that this manuscript has done some metabolite analysis, it is better to change the title of this manuscript and highlight metabolite part, such as integrating multi-omics data.

The authors included too many descriptions to show the significant "improvements" to the published data, Sections "Genome evolution of typical herb angiosperm plants", "Inconsistent phylogenetic tree and differentiation time estimation in the published *A. sinensis* reference genome" and "comprehensive comparisons of published data with Qinggui1" can be reorganized and shortened, and some text should be moved into "methods" and "discussion" section.

The authors provide the SNP between *A. sinensis* (GS) and *A. sinensis* (QH), are there any metabolites difference? are these SNPs associated with the metabolites? It would be great if the authors can provide some metabolite related examples by using targeted or untargeted analysis.

Line 99-108, metabolite profiling and transcriptome study also need highlighted here.

Line 33-34, after reading the manuscript, the description of "the detailed pathways and metabolic genes....." is overstated.

Line 135-136, the authors stated that 80.24% of the assembly was identified to be repetitive sequence and the Apiaceae family coriander genome is 70.59%, 10% of difference is not "slightly higher", is actually "higher".

Line 201, reword the sentence "this article believed that...."

Reviewer #3 (Remarks to the Author):

The manuscript titled "The genome of *Angelica sinensis* provides new insights into the evolution and production of medicinal compounds" offers valuable genomic data on the medicinal plant *Angelica sinensis*. The genome of *A. sinensis* cultivar Qinggui1 (QH) genome, an Apiaceae family member, was sequenced using PacBio SMRT sequences, paired HiSeq reads, and Hi-C reads. The final assembly resulted in a genome size of 2.16 Gb. This assembly was compared to the previously available *A. sinensis* genome from Gansu province (Han et al. 2022), which was 2.37 Gb. The authors also investigate the difference of transcriptome and metabolites between early and late flowering ecotypes, as early flowering significantly reduces the amount of important essential oils composed of phthalides in roots, offering valuable understanding regarding biosynthetic pathways of important medicinal compounds.

The overall study presents valuable insights and contributes significantly to the field, making it suitable for publication. However, the manuscript needs thorough proofreading to address several spelling mistakes and improve sentence organization. I have provided feedback on the WGD and comparative genomics section, but there are also spelling mistakes in other sections that require careful revision. Engaging a native speaker or professional proofreader could greatly assist in rectifying these language-related concerns. By carefully attending to these issues, the manuscript can fulfill its potential and effectively convey its important findings to the scientific community.

Major comments:

L213 – L255: The section is written in a complex and convoluted manner, making it difficult to follow the main points. Especially the explanations regarding the peak charts and Ks values are confusing and lack clarity. The organization of information is unclear, jumping between different comparisons and analyses without clear transitions. The comparisons between different genomes and species could be better structured and presented more coherently.

L228-L240: It is unclear whether it discusses the results of Han et al. or the current publication. To enhance clarity, it would be better to present the analysis from the current study first and then compare it to the findings of Han et al.

Furthermore, in L234: "it can be seen that...", it is not specified where to find the information mentioned. The appropriate figure number should be provided for clarity.

L234-L237: This section discusses the expected nature of Ks peaks within and between species. It would be beneficial to explain that Ks peaks within species occur due to the comparison of paralogs resulting from whole-genome duplication events, while Ks peaks between species indicate the divergence of orthologs.

L237-L238: Please give appropriate reference to the general belief.

In several places in this section, the authors discuss the order of the different peaks without performing any statistical test. Overall, the section would benefit from simplified language, robust statistics, clearer explanations, and better transitions between ideas.

Reviewers' comments:

Reviewer #1 (Remarks to the Author):

The article "The genome of *Angelica sinensis* provides new insights into the evolution and production of medicinal compounds" by Li et al., presents an improved genome for *Angelica sinensis* var. *Qinggui1* (*Angelica* roots), a Chinese traditional medicinal herb. The researchers decoded the genome, identified pathways and genes responsible for producing bioactive components in *Angelica* roots, and employed a multi-omics approach including metabolomic profiling and transcriptome analysis. The study revealed expansions of key genes involved in the biosynthesis of bioactive compounds, offering valuable insights for future synthetic biology and herb breeding research.

Before considering publication, it is crucial to address the numerous questions raised in the manuscript regarding the experiments and analysis. One important aspect to consider is clearly describing the differences between the previous analysis/publication and the current study, along with providing a rationale for the changes made. Additionally, a more detailed comparison is needed instead of solely focusing on the numerical differences.

Response: Thank you for your suggestion. In the revised manuscript, we further provide more details and discussions on the comparison of two genome assemblies in the Discussion section (Line 592-607).

For instance, the authors have addressed a critical limitation in a previous publication by utilizing a more comprehensive approach that involved a larger number of genes. While they emphasize the advantage of comparing over 2,000 one-to-one orthologous genes in their study (Line 198-211), it remains uncertain whether this increased gene count directly translates to greater accuracy in their results. It would be valuable if the authors could provide additional criteria or evidence supporting the claim that their approach yields more accurate results compared to the previously published article, beyond the mere increase in the number of genes compared.

Additionally, enhancing the manuscript can be accomplished by addressing concerns such as providing a more comprehensive explanation of the methodology utilized and presenting a clearer justification for the research undertaken.

Response: Thank you for your kind suggestion. We have provided additional supplemental Data to improve the understanding of our analysis and methods.

For a long time, plant phylogeny has been analyzed using plastid genes, mitochondrial genes or a few conserved single-copy nuclear genes. In this study, we collected nuclear gene sets of 20 species from annotated genomes. By using a clustering and phylogenetic-based ortholog inference method (Yang and Smith, 2014), we obtained a total of 10,398 one-to-one orthologous gene families (one sequence per species for each orthologous gene family), 2,133 of which were present in all 20 species. **This same or similar method had been utilized in many previous publications** (Yang et al., 2015; Hui et al., 2017; NEVADO, 2016; JOSEPH, 2017; MENG, 2018), which suggested that this approach was a good method. Of course, we could not claim that the increased gene count can directly translate to greater accurate. Actually, the authors constructed a lot of species trees in these publications, using different gene number and compared them, but **it could not prove** the greater accurate of increased gene number.

The accuracy of evolutionary tree construction depends on **many factors** and is thus **very complex**. Firstly, it is the type and size of the data set. Not only the morphological traits data, plastid genes, mitochondrial genes and nucleogenetic sequences were used to build different progressive trees (Endress & Doyle, 2009; Soltis et al., 2011; Ruhfel et al., 2014; Zeng et al., 2014), but also

progression trees constructed using full-length nucleic acid sequences or using only nucleic acid and amino acid sequences at a specific site of the gene codon are different (Wickett et al., 2014). Secondly, it is the method and model of constructing tree. The methods have concatenation and coalescence. The concatenation method is to construct a phylogenetic tree using software RAxML (Stamatakis, 2014) or iqtree (Nguyen et al., 2015) by concatenating all genes into one whole body. The coalescence method is to first build a tree for each gene and then use the software ASTRAL (Zhang et al., 2017) to build a common tree for all gene trees (Wickett et al., 2017). The models used to construct phylogenetic trees are more diverse, such as nuclear acid model GTR, HKY, JC, F81, K2P, K3P, K81uf and protein model WAG, LG, Poisson, cpREV, mtREV, Dayhoff, mtMAM, JTT (Nguyen et al., 2015). Therefore, it nearly could not prove that the mere increased gene count can directly translate to greater accurate.

In this study, with the 2,133 one-to-one orthologous gene families, we applied both coalescent and concatenation methods to reconstruct the phylogeny using these gene dataset. The topology of **concatenation method tree was better**, because the loci of *Arabidopsis thaliana* and *Populus trichocarpa* are the same with the APG IV angiosperm phylogeny.

All by all, we could not opinionated claim that a published method can be greater accurate. There are many methods to construct species trees. We only used the approach that many people have utilized a lot recently and **not just use single-copy genes. Generally speaking**, with the increase of species number, **we only obtain very little single-copy genes in all species**, so we used the new approach to obtain more genes to construct phylogeny tree due to as many as 20 species in this study (Actually, we nearly could not obtain single-copy genes existing in all 20 species).

Reference

- YANG Y, SMITH SA, 2014. Orthology inference in nonmodel organisms using transcriptomes and low-coverage genomes: improving accuracy and matrix occupancy for phylogenomics[J]. *Molecular biology and evolution*, 31(11): 3081-3092.
- YANG Y, MOORE MJ, SAMUEL FB et al. (2015). Dissecting molecular evolution in the highly diverse plant clade caryophyllales using transcriptome sequencing[J]. *Molecular Biology & Evolution*(8), 2001-2014.
- HUI S, DONGMEI J, JIANGPING S, et al. Large scale phylogenomic analysis resolves a backbone phylogeny in ferns[J]. *Gigascience*, 7(2).
- NEVADO B, ATCHISON GW, HUGHES CE et al. Widespread adaptive evolution during repeated evolutionary radiations in New World lupins. *Nat Commun*[J]. 2016 Aug 8;7:12384.
- JOSEPH F, WALKER, Yang, et al. (2017). Widespread paleopolyploidy, gene tree conflict, and recalcitrant relationships among the carnivorous caryophyllales. *American journal of botany*[J].
- MENG W, KOSTYUN JL , HAHN MW et al. (2018). Dissecting the basis of novel trait evolution in a radiation with widespread phylogenetic discordance. *Molecular Ecology*[J]. 27.
- ENDRESS PK, DOYLE JA, 2009. Reconstructing the ancestral angiosperm flower and its initial specializations[J]. *American journal of botany*, 96(1): 22-66.
- SOLTIS DE, SMITH SA, CELLINESE N, et al., 2011. Angiosperm phylogeny: 17 genes, 640 taxa[J]. *American journal of botany*, 98(4): 704-730.
- RUHFEL BR, GITZENDANNER MA, SOLTIS PS, et al., 2014. From algae to angiosperms—inferring the phylogeny of green plants (Viridiplantae) from 360 plastid genomes[J]. *BMC Evolutionary Biology*, 14(1): 23.
- ZENG LP, ZHANG Q, SUN RR, et al., 2014. Resolution of deep angiosperm phylogeny using conserved nuclear genes and estimates of early divergence times[J]. *Nature communications*, 5(1): 4956.
- WICKETT NJ, MIRARAB S, NGUYEN N, et al., 2014. Phylotranscriptomic analysis of the origin and early diversification of land plants[J]. *Proceedings of the National Academy of Sciences of the United States of America*, 111(45): 4859-4868.
- STAMATAKIS A, 2014. RAxML Version 8: A tool for Phylogenetic Analysis and Post-Analysis of Large Phylogenies[J]. *Bioinformatics*, 30(9): 1312-1313.
- NGUYEN LT, SCHMIDT HA, VON HAESLER A, et al., 2015. IQ-TREE: a fast and effective stochastic algorithm for estimating maximum-likelihood phylogenies[J]. *Molecular biology and evolution*, 32(1): 268-274.
- ZHANG C, SAYYARI E, MIRARAB S, 2017. ASTRAL-III: Increased Scalability and Impacts of Contracting Low Support Branches[J]. *RECOMB International Workshop on Comparative*

Genomics, Springer, Cham: 53-75.

Further comments are listed below.

Subtitle

It would be necessary to include clear and informative subtitles throughout the paper, particularly in parts where the current subtitle lacks clarity. This will help readers grasp the main points of each paragraph more effectively. e.g. In line 195, 196 "Inconsistent phylogenetic tree and differentiation time estimation in the published *A. sinensis* reference genome". I firmly believe that a more appropriate title can be selected for this section.

Response: Thank you for your kind suggestion. We reorganized the paper structure and revised the text, especially the title and subtitles (line 167-168).

Phylogenetic analysis

The manuscript discusses the selection of 20 representative angiosperms for phylogenetic analysis. However, the criteria for choosing these specific species are not adequately explained. It will enhance the manuscript if the author provides more detailed information regarding the 'selection process'. Given the abundance of important medicinal species, it is important to clarify why certain species were chosen and whether other significant.

Response: Thanks. In the Results section we have added the rationale and reasons for the selection of these species (Line 180-198).

Plant preparation and DNA sequencing

I suggest the author provide clearer details regarding the 'age' and 'timing of plant collection' for sequencing. The phrase "same place, same time, right time" is vague and could be improved by specifying the specific age of the plants at the time of sample collection. Additionally, it would be necessary for author to include information about the growth conditions such as the light cycle, humidity, and age of the plants at the time of sample collection.

Response: Thank you for your suggestion. We provided a more detailed introduction to the materials methods in the supplementary data.

Citations

The author should consider including appropriate citations where necessary to support their statements. For instance, in line 254, when referring to "Han's article", author should cite Han's article in this section. It is one example, and I found several places author need to include appropriate citations.

Response: We carefully checked and added the references.

Line 264

It would be helpful to specify which common databases are used for this particular part of the study.

Response: We marked the name of the specific database (Line 296-298).

Line 637, 646

The author introduced key metabolite genes, and I believe that it would be necessary for

authors to provide more descriptive information to aid the understanding of readers who are unfamiliar with the metabolic pathways. Consider including figures or tables to visually illustrate the pathways and provide further context, instead of presenting a list without additional explanation. This would enhance the clarity and accessibility of this section

Response: Thanks. We described some of the genes with important functions with details (Line 643-644, Line 652).

Line 745

There is more than one space between S18 and S19.

Response: We carefully revised the full text to avoid similar errors.

Reviewer #2 (Remarks to the Author):

This study generated the chromosome scale genome of *A. sinensis*. The study also provides a large set of omics data for root in *A. sinensis*. Since a genome sequence manuscript has already been published in "the Plant Journal", in this study, the authors have performed comprehensive comparisons with the published data and claimed it is an improved chromosome-level genome for *A. sinensis*. However, the data presentation still needs lots of improvement to fully support the conclusion. The following suggestions may help to improve the manuscript, which is encouraged to resubmit with revisions.

The major difference of the two manuscripts is that this manuscript has done some metabolite analysis, it is better to change the title of this manuscript and highlight metabolite part, such as integrating multi-omics data.

Response: As suggested, we changed the title to be "Integrating the genomic and multiomic data of *Angelica sinensis* provides new insights into the evolution and biosynthesis of pharmaceutically bioactive compounds".

The authors included too many descriptions to show the significant "improvements" to the published data, Sections "Genome evolution of typical herb angiosperm plants", "Inconsistent phylogenetic tree and differentiation time estimation in the published *A. sinensis* reference genome" and "comprehensive comparisons of published data with Qinggui1" can be reorganized and shortened, and some text should be moved into "methods" and "discussion" section.

Response: Thank you for your suggestion. We have reorganized and merged this part of the content according to your suggestion (Line 170-288). And in the discussion section, the improvement of the genome is specially discussed (Line 592-607).

The authors provide the SNP between *A. sinensis* (GS) and *A. sinensis* (QH), are there any metabolites difference? are these SNPs associated with the metabolites? It would be great if the authors can provide some metabolite related examples by using targeted or untargeted analysis.

Response: Thank you for your suggestion. Due to the limitations of samples and materials, it is not easy to perform metabolome sequencing and make comparisons. For the genes related to anabolic metabolism affected by these SNPs, we performed a sorting and enrichment analysis, please see Table S11, Figure 3d.

Line 99-108, metabolite profiling and transcriptome study also need highlighted here.

Response: Thanks. We revised the paragraph following your suggestion (Line 109-110).

Line 33-34, after reading the manuscript, the description of "the detailed pathways and metabolic genes....." is overstated.

Response: As suggested, we modified this sentence to "we deciphered the pathways and critical metabolic genes for" (Line 38).

Line 135-136, the authors stated that 80.24% of the assembly was identified to be repetitive sequence and the Apiaceae family coriander genome is 70.59%, 10% of difference is not "slightly higher", is actually "higher".

Response: Revised (Line 139).

Line 201, reword the sentence “this article believed that....”

Response: We deleted this sentence.

Reviewer #3 (Remarks to the Author):

The manuscript titled "The genome of *Angelica sinensis* provides new insights into the evolution and production of medicinal compounds" offers valuable genomic on the medicinal plant *Angelica sinensis*. The genome of *A. sinensis* cultivar Qinggui1 (QH) genome, an Apiaceae family member, was sequenced using PacBio SMRT sequences, paired HiSeq reads, and Hi-C reads. The final assembly resulted in genome size of 2.16 Gb. This assembly was compared to the previously available *A. sinensis* genome from Gansu province (Han et al. 2022), which was 2.37 Gb. The authors also investigate the difference of transcriptome and metabolites between early and late flowering ecotypes, as early flowering significantly reduces the amount of important essential oils composed of phthalides in roots, offering valuable understanding regarding biosynthetic pathways of important medicinal compounds.

The overall study presents valuable insights and contributes significantly to the field, making it suitable for publication. However, the manuscript needs thorough proofreading to address several spelling mistakes and improve sentence organization. I have provided feedback on the WGD and comparative genomics section, but there are also spelling mistakes in other sections that require careful revision. Engaging a native speaker or professional proofreader could greatly assist in rectifying these language-related concerns. By carefully attending to these issues, the manuscript can fulfill its potential and effectively convey its important findings to the scientific community.

Response: Thank you for your comments. We have invited some experts with English as native language to improve the writings.

Major comments:

L213 – L255: The section is written in a complex and convoluted manner, making it difficult to follow the main points. Especially the explanations regarding the peak charts and Ks values are confusing and lack clarity. The organization of information is unclear, jumping between different comparisons and analyses without clear transitions. The comparisons between different genomes and species could be better structured and presented more coherently.

Response: We have rewritten these parts to make the contents clear according to your suggestion (Line 251-288).

L228-L240: It is unclear whether it discusses the results of Han et al. or the current publication. To enhance clarity, it would be better to present the analysis from the current study first and then compare it to the findings of Han et al.

Response: Thank you for your suggestion. In the revised manuscript, we clearly distinguish whether the results were obtained in our analyses or just from the published paper. The comparison with Han et al has been moved to the Discussion section (Line 609-625).

Furthermore, in L234: "it can be seen that...", it is not specified where to find the information mentioned. The appropriate figure number should be provided for clarity.

Response: We have revised this paragraph (Line 273-288).

L234-L237: This section discusses the expected nature of Ks peaks within and between

species. It would be beneficial to explain that Ks peaks within species occur due to the comparison of paralogs resulting from whole-genome duplication events, while Ks peaks between species indicate the divergence of orthologs.

Response: We have rewritten this paragraph as your suggestion (Line 273-288).

L237-L238: Please give appropriate reference to the general belief.

Response: Revised.

In several places in this section, the authors discuss the order of the different peaks without performing any statistical test.

Response: Thank you for your suggestion. Sorry that too few data points to support a suitable statistical method for testing. Instead, we listed the peak values in detail for a more intuitive comparison in the revised manuscript (Line 277-278, Line 283-284, Table S10).

Overall, the section would benefit from simplified language, robust statistics, clearer explanations, and better transitions between ideas.

Response: Thank you for your suggestion. We have carefully modified the manuscript and invited native speakers to polish the text, believing that the writing quality has been greatly improved.

Reviewers' comments:

Reviewer #1 (Remarks to the Author):

After reviewing the author's rebuttal letter addressing my comments and those of the other reviewers, I found myself mostly convinced by their edits. While there were still a few aspects where I couldn't fully grasp the significance of this research compared to the previous work, I believe that, on the whole, this study holds clear importance in the field. I don't have any further comments to add regarding their research.

Reviewer #2 (Remarks to the Author):

The authors have addressed all my comments and concerns, I have no more question and it can be accepted for publication.

Reviewer #4 (Remarks to the Author):

Review 3 mentions the problem of WGD events, the authors did not compare with previous studies, and the KS analysis given by the authors only has two peaks, which is different from the three peaks of the previously published genome.

Review 3 mentions the problem of WGD events, the authors did not compare with previous studies, and the KS analysis given by the authors only has two peaks, which is different from the three peaks of the previously published genome.

We appreciate your suggestion. Ancient Whole Genome Duplications (WGDs) are typically detected by examining the distribution of synonymous substitutions per site (Ks) among paralogous genes within a genome, which can be visualized in a “Ks plot”¹⁻³. In the absence of WGDs or large episodic duplications, the synonymous substitutions between paralogs within a genome should follow an exponential distribution⁴. WGDs should produce additional normally distributed peaks in the Ks plots^{5,6}.

In the study by Han et al.⁷, the Ks plot displays three peaks, located at approximately 0.5, 1.0, and 1.7 (Figure 1). As suggested by Paterson et al.⁸ and Conant et al.⁹, if a WGD event is too ancient, it may not be detectable using a Ks plot due to the loss of too many duplicate genes from the ancient WGD. Theoretically, the older the WGD event, the more duplicate genes are lost, resulting in a weaker Ks signal. In Han et al.’s study, the peak at around 1.7 is unusually high, even higher than the peaks at around 0.5 and 1.0, which seems unreasonable.

In contrast, our results show very obvious peaks at 0.5 and 1.0, but the peak around 1.7 is not prominent (Figure 2), we thus ignore this older WGD event. Our result is also consistent with the previous study¹⁰ while we observe the Ks plot of grape which is used as the model organism (Figure 3), and WGT- γ is absent in most other plant genomes. We recommend maintaining a balanced emphasis in the manuscript to avoid giving an impression of being overly critical.

Han et al.'s paper provides only a concise description of their Ks analysis methods, making it challenging to compare specific methods and parameters with those used in our study. Our analysis method is described as follows: First, we used diamond¹¹ (v2.1.6.160) to compare (-e 1e-5) the gene set of the selected species to obtain blast comparison information. Second, we combined this information with gff annotation file information to perform collinearity analysis using the wgdi collinearity module (-icl). Third, we estimated synonymous substitutions per synonymous site (Ks) between collinear genes using the yn00 program as implemented in the PAML package¹². Finally, we illustrated Ks distribution of orthologous blocks using WGDI toolkit¹³ and the median Ks values of homologous genes were used to classify blocks generated by each duplication event.

Figure 1 K_s distribution from orthologs and paralogs among five Apiaceae species. (Figure 2b in Han et al.'s study)

Figure 2 The K_s distribution for orthologous gene pairs within and between Apiaceae species. (Figure 2e in our study)

Figure 3 The distribution of Ks for WGD gene pairs from kiwifruit, tea plant, rhododendron, and grape. (Figure 2d in Wang, Y. et al.'s study)

- 1 Cui, L. *et al.* Widespread genome duplications throughout the history of flowering plants. *Genome research* **16**, 738-749 (2006). <https://doi.org:10.1101/gr.4825606>
- 2 Barker, M. S. *et al.* Multiple paleopolyploidizations during the evolution of the Compositae reveal parallel patterns of duplicate gene retention after millions of years. *Molecular biology and evolution* **25**, 2445-2455 (2008). <https://doi.org:10.1093/molbev/msn187>
- 3 Vanneste, K., Baele, G., Maere, S. & Van de Peer, Y. Analysis of 41 plant genomes supports a wave of successful genome duplications in association with the Cretaceous-Paleogene boundary. *Genome research* **24**, 1334-1347 (2014). <https://doi.org:10.1101/gr.168997.113>
- 4 Lynch, M. & Conery, J. S. The evolutionary demography of duplicate genes. *Journal of Structural and Functional Genomics* **3**, 35-44 (2003). <https://doi.org:10.1023/A:1022696612931>

- 5 Blanc, G. & Wolfe, K. H. Widespread paleopolyploidy in model plant species
inferred from age distributions of duplicate genes. *The Plant cell* **16**, 1667–
1678 (2004). <https://doi.org/10.1105/tpc.021345>
- 6 Schlueter, J. A. *et al.* Mining EST databases to resolve evolutionary events
in major crop species. *Genome* **47**, 868–876 (2004).
<https://doi.org/10.1139/g04-047>
- 7 Han, X. *et al.* The chromosome-level genome of female ginseng (*Angelica
sinensis*) provides insights into molecular mechanisms and evolution of
coumarin biosynthesis. *The Plant journal : for cell and molecular biology*
112, 1224–1237 (2022). <https://doi.org/10.1111/tpj.16007>
- 8 Paterson, A. H., Bowers, J. E. & Chapman, B. A. Ancient polyploidization
predating divergence of the cereals, and its consequences for comparative
genomics. *Proceedings of the National Academy of Sciences of the United States
of America* **101**, 9903–9908 (2004). <https://doi.org/10.1073/pnas.0307901101>
- 9 Conant, G. C., Birchler, J. A. & Pires, J. C. Dosage, duplication, and
diploidization: clarifying the interplay of multiple models for duplicate
gene evolution over time. *Curr Opin Plant Biol* **19**, 91–98 (2014).
<https://doi.org/10.1016/j.pbi.2014.05.008>
- 10 Wang, Y. *et al.* An ancient whole-genome duplication event and its contribution
to flavor compounds in the tea plant (*Camellia sinensis*). *Horticulture
research* **8**, 176 (2021). <https://doi.org/10.1038/s41438-021-00613-z>
- 11 Buchfink, B., Reuter, K. & Drost, H. G. Sensitive protein alignments at tree-
of-life scale using DIAMOND. *Nature methods* **18**, 366–368 (2021).
<https://doi.org/10.1038/s41592-021-01101-x>
- 12 Yang, Z. PAML 4: phylogenetic analysis by maximum likelihood. *Molecular
biology and evolution* **24**, 1586–1591 (2007).
<https://doi.org/10.1093/molbev/msm088>
- 13 Sun, P. *et al.* WGDI: A user-friendly toolkit for evolutionary analyses of
whole-genome duplications and ancestral karyotypes. *Mol Plant* **15**, 1841–1851
(2022). <https://doi.org/10.1016/j.molp.2022.10.018>

Reviewers' comments:

Reviewer #4 (Remarks to the Author):

1. The D and E parts of fig2 look confused, is the ancient wgd in Apiaceae the WGT of the core eudicot ancestor or what? And the upper and lower parts are not the same.

The lower part of Fig. 2 d and e is best not to be, it seems unclear.

2. The KS peaks of the ancestors of core eudicots will only have if they are fitted, not that they cannot be drawn.

Please refer to this article,

The chromosome-level genome of female ginseng (*Angelica sinensis*) provides insights into molecular mechanisms and evolution of coumarin biosynthesis.

A. sinensis has experienced three polyploidy events, one of the core eudicots ancestors, and then two recent occurrences.

That is, the ancient wgd in Apiaceae marked in the fig2 D and E, is actually the WGT event of the core eudicot ancestor. Please revise this.

3. Line 182, dicotyledonous, is generally written as eudicots.

Reviewers' comments:

Reviewer #4 (Remarks to the Author):

1. The D and E parts of fig2 look confused, is the ancient wgd in Apiaceae the WGT of the core eudicot ancestor or what? And the upper and lower parts are not the same. The lower part of Fig. 2 d and e is best not to be, it seems unclear.

Response: Thank you for your suggestion. As mentioned by Reviewer #3, Ks peaks within species occur due to the comparison of paralogs resulting from whole-genome duplication events, while Ks peaks between species indicate the divergence of orthologs. In light of your comments and those made by Reviewer #3, we have decided to remove the lower part of Figure 2 (parts d and e) to avoid any potential confusion. We have also revised the corresponding sections of the manuscript text accordingly. We believe that these changes will enhance the clarity of our work and help readers better understand our findings.

2. The KS peaks of the ancestors of core eudicots will only have if they are fitted, not that they cannot be drawn.

Please refer to this article,

The chromosome-level genome of female ginseng (*Angelica sinensis*) provides insights into molecular mechanisms and evolution of coumarin biosynthesis.

A. sinensis has experienced three polyploidy events, one of the core eudicots ancestors, and then two recent occurrences.

That is, the ancient wgd in Apiaceae marked in the fig2 D and E, is actually the WGT event of the core eudicot ancestor. Please revise this.

Response: Thank you for your suggestion. We have made revisions to the sections pertaining to Figure 2d and 2e (Lines 268-290). We have placed greater emphasis on the fact that *A. sinensis* has undergone three polyploidy events, as supported by the referenced article (Line 274). We appreciate your feedback and believe these changes enhance the clarity and accuracy of our manuscript.

3. Line 182, dicotyledonous, is generally written as eudicots.

Response: Thank you for your suggestion. We have revised dicotyledonous as eudicot plants. (Line 182)

REVIEWERS' COMMENTS:

Reviewer #4 (Remarks to the Author):

I have no more questions, and I am satisfied with the author's reply.